# CausalChaos! Dataset for Comprehensive Causal Action Question Answering Over Longer Causal Chains Grounded in Dynamic Visual Scenes

**Paritosh Parmar**[1,2]**, Eric Peh**[1,2]**, Ruirui Chen**[1]**, Ting En Lam**[3]**,**
**Yuhan Chen**[4]**, Elston Tan**[5]**, Basura Fernando**[1,2]

[1]Institute of High Performance Computing, Agency for Science, Technology and Research, Singapore
[2]Centre for Frontier AI Research, Agency for Science, Technology and Research, Singapore
[3]Nanyang Technological University, [4]National University of Singapore, [5]Singapore Polytechnic
https://github.com/LUNAProject22/CausalChaos

## Abstract

Causal video question answering (QA) has garnered increasing interest, yet existing datasets often lack depth in causal reasoning. To address this gap, we capitalize on the unique properties of cartoons and construct CausalChaos!, a novel, challenging causal Why-QA dataset built upon the iconic "Tom and Jerry" cartoon series. Cartoons use the principles of animation that allow animators to create expressive, unambiguous causal relationships between events to form a coherent storyline. Utilizing these properties, along with thought-provoking questions and multi-level answers (answer and detailed causal explanation), our questions involve causal chains that interconnect multiple dynamic interactions between characters and visual scenes. These factors demand models to solve more challenging, yet well-defined causal relationships. We also introduce hard incorrect answer mining, including a causally confusing version that is even more challenging. While models perform well, there is much room for improvement, especially, on open-ended answers. We identify more advanced/explicit causal relationship modeling & joint modeling of vision and language as the immediate areas for future efforts to focus upon. Along with the other complementary datasets, our new challenging dataset will pave the way for these developments in the field.

## 1 Introduction

Understanding the intricate *motivations* behind human actions is paramount in developing sophisticated systems capable of nuanced behavior analysis. In real-world scenarios, actions are shaped by a *multitude of factors*, including personal experiences, emotions, social contexts, and cultural backgrounds. This complexity necessitates a *comprehensive* approach to unraveling the "*why*" behind actions, fostering empathy, effective communication, and robust decision-making. *Causal video question answering* aims to decipher the answers behind characters' actions. Despite the growing interest in causal video QA, existing datasets often fall short, requiring only: **1)** *surface-level understanding*; or **2)** *involve more of simple word substitution in the QA pairs, rather than causal reasoning* (*e.g.*, "Q. Why are the cars on the street not moving? A. Parked."). Recognizing this gap, we embark on the development of a rigorous and challenging *causal Video-QA dataset*. Our goal is to provide a high-quality resource that rigorously evaluates and advances causal video QA models.

Drawing on the established benefits of cartoons with cognitive processes among children [40, 41, 31, 55, 42], we leverage the renowned series **TOM~JERRY** to *create a novel and demanding causal Why-QA dataset* called **CausalChaos!**. The key characteristics of CausalChaos! are as follows:

38th Conference on Neural Information Processing Systems (NeurIPS 2024) Track on Datasets and Benchmarks.

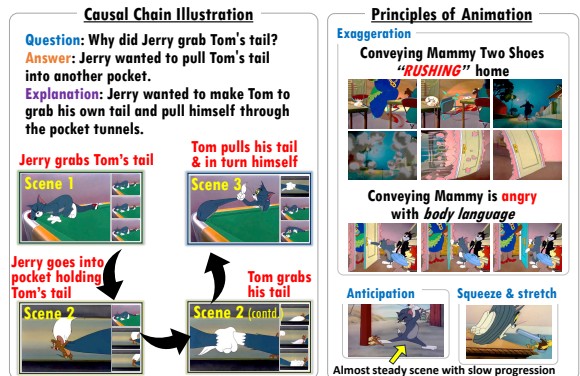

Figure 1: **(Left) Examples of causal questions about characters' actions from our CausalChaos! dataset— based on Tom & Jerry cartoon series.** Q: question; A: answer; E: explanation. *Please view in Adobe Reader to play the embedded videos for better explanation.* **(Middle) Illustration of causal chain, scene changes.** Linking multiple clues/cues embedded in different scenes to solve causal relationships pose a challenge for VideoQA models. **(Right)** Animators leverage **Principles of Animation** to *stylize* the *visuals* & *motions* to *disentangle/highlight key content* of the scene to create *well-defined/unambiguous and effectively communicated cause-and-effect relationships*. The interplay of these factors allow models to focus on solving complex, yet, well-defined, unambiguous causal relationships. We have provided enlarged figures in Appendix.

- As demonstrated in Figure 1, 2, we formulate *thought-provoking questions* (**Q**), where the *answer* (**A**) and its *detailed explanation* (**E**) aim to enhance the model's understanding of the causal chains of visual events in a given video clip. A causal chain is a sequence of events/actions in which each step influences or leads to the next, creating a cause-and-effect relationship. It illustrates how one event/action produces a subsequent event/outcome. Compared to existing datasets, CausalChaos! presents **longer causal chains**, as highlighted in Figure 2 and discussed in section 2.
- Video clips in our dataset feature frequent **scene** and **shot changes**. Here, a scene change or shot change refers to the transition from one visual setting or perspective to another. It typically involves shifting focus to a new location, action, or character within the storyline. The *links* and *causes of the causal chains are often dispersed across various scenes*. Consequently, models are challenged to exert greater cognitive effort in *connecting multiple events* (scenes) and identifying intermediate causes to comprehend the "why" questions related to the clips.
- Despite the complexity and length of the causal chains, they are distinctly *delineated*, **unambiguous**, and **effectively communicated** using *principles of animation* [44] like staging and exaggeration. This deliberate design allows models to *focus on deciphering causal relationships*.
- CausalChaos! introduces an added layer of complexity by necessitating the modeling of actions at various levels of granularity—ranging from sweeping, **large-scale movements** to nuanced, **finegrained actions**, such as interpreting emotional cues through facial expressions.
- Our dataset demands a **diverse range of reasoning skills**, encompassing deductive, spatial, emotional reasoning, and more, as outlined in Figure 2 and discussed in section 3.
- We introduce **challenging incorrect options**, including the CausalConfusion set, to prevent models from relying on shortcuts, such as object-noun or action-verb matching in vision-language spaces, and instead require them to understand causal relationships.

Upon evaluating various state-of-the-art VideoQA models including the recent multimodal instruction tuning models, we found that our dataset remains one of the most challenging causal QA dataset. Particularly, we observed that models often: **1)** *jumped to conclusions* based on *partial evidence*, rather than considering the full set of evidence; **2)** *failed to engage in true causal reasoning*, opting instead for shortcuts like object/action-noun/verb matching to select answers. Based on this, we identified *more advanced/explicit causal relationship modeling and jointly modeling vision and language* as the immediate areas for future efforts to focus upon. Further, we show that similar to how cartoons help children better connect cause and effects, they can help VideoQA models as well. We *incorporated our dataset with the NextQA* [49], a *real-world dataset* and found that it *brings some improvements in why questions*. What is more, *incorporating our dataset* brings improvement on *non-Why questions* as well.

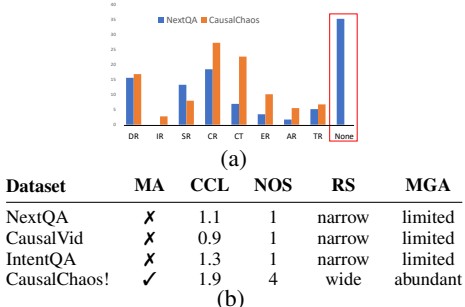

(a)

| Dataset | MA | CCL | NOS | RS | MGA |
|---|---|---|---|---|---|
| NextQA | ✗ | 1.1 | 1 | narrow | limited |
| CausalVid | ✗ | 0.9 | 1 | narrow | limited |
| IntentQA | ✗ | 1.3 | 1 | narrow | limited |
| CausalChaos! | ✓ | 1.9 | 4 | wide | abundant |

(b)                                                                    (c)

Figure 2: **(a) Types of reasoning demanded by our CausalChaos! dataset.** Reasoning types: DR-*deductive* reasoning; IR-*inductive*; SR-*spatial*; CR-*causal*; CT-*critical thinking*; ER-*emotional*; AR-*abductive*; TR-*temporal*; None-no reasoning required as per the human subjects. None is undesirable, and tend to indicate that questions are less challenging. **(b) Comparison among CausalChaos! and existing causal videoQA datasets.** MA-multilevel answers; CCL-causal chain length; NOS-no. of scenes; RS-reasoning spectrum; MGA-multigranular actions. **(c) Qualitative comparison between CausalChaos! and NextQA (Why-QA) datasets.** CausalChaos! Answers and Explanations give detailed information regarding the actual cause-and-effect relationships, motivations, emotions covering wide range of reasoning types. Note that, we have temporally cropped videos to retain only the relevant parts from NextQA dataset videos; otherwise, raw videos are longer resulting in unintended problem of temporal localization for models.

Overall, we believe our CausalChaos! dataset presents challenges spanning the entire VideoQA pipeline, from *deciphering intricate videos* to *processing complex questions* and *discerning nuanced answers*, stimulating research in *video processing*, *causal reasoning*, *language modeling*, and *joint modeling*. Along with the other complementary datasets, such as [49, 17, 19, 21, 48], our new challenging dataset will pave the way for these developments in the field.

## 2  Related Work

To drive the progress in VideoQA, researchers have developed various datasets [27, 50, 10, 60, 58, 63, 13, 14, 56, 57, 9, 48] with distinct focuses and contributions—see the survey [61]. In the following, we mainly compare our work with prior art focusing on causal QA literature. We provide a detailed discussion on various VideoQA datasets and models in the Appendix. Our work is inspired from a few datasets: *CLEVRER* [54, 29], *NextQA* [49], *CausalVidQA* [17], *IntentQA* [19] that cover causal reasoning. However, they have limitations that our dataset aims to fill:

- *Limitation in scope*: *e.g.*, [54, 29] only consider *collision events of simple inanimate objects* such as cuboids; as a result, *lack: actors*, their *characteristics* such as *emotions*, *intentions*. Our dataset has actors whose actions are shaped by *emotions*, *intentions*, *context*, etc.
- *Lack of precise temporal annotations* (e.g., [49, 17]) *inadvertently* involves *temporal localization* & may create a *false sense of difficulty*, while the questions may *not be challenging*. We alleviate this problem by providing precise temporal annotations and focusing on designing *truly complex* and challenging QA.
- *Lack of complexity in questions* (*e.g.*[49, 17, 19]). Based on human studies, we found that a majority of causal QA in these datasets were flagged as *not requiring any notable reasoning* (Figure 2(a)). We also attempted to *quantify* the *complexity in reasoning* by computing the *lengths of causal chains* involved in their QAs vs. ours using GPT-4o [34]. Causal chains in [49, 17, 19] are shorter (Figure 2(b). In comparison, our dataset has *longer causal chains*—average length 1.9—posing a challenge for models in *connecting multiple events* or cues. Our dataset also demands *wide spectrum of types of reasoning* (Figure 2a).
- Scenes in [54, 29, 49, 17, 19] are *less dynamic*, mostly involving a *single scene*. In comparison, our dataset averages about *four scenes*. *Rapid scene changes* & *dynamic interactions* challenge VideoQA models to link *context* & *cues across different scenes*, modeling cause-and-effect relationships over *longer causal chains* (more details in subsection 3.2). While similar challenges have been acknowledged in other computer vision problems [36], they remain unexplored in VideoQA.

- *Lack of hard negatives.* Current datasets may not emphasize on including hard incorrect options. Due to this, models may not be required to do causal reasoning, but rather they can exploit *shortcuts like object/action-noun/verb matching* in vision-language spaces to select correct answer. To address this limitation, we develop *hard negative selection strategies*.
- *No hierarchical answers with different level of explanation.* These datasets contain a *single-level answer to 'Why'*-questions. On the other hand, we richly annotate our dataset to provide *two-level answers to 'Why'-questions*—1) direct/immediate cause; 2) deeper explanation. Quality of answers is further enhanced, and are *more informative* as a consequence of the *more complex questions*.

## 3 CausalChaos! Dataset

This section details the construction of our VideoQA dataset designed to challenge causal reasoning, covering its *video source*, *annotation process*, & *quality checks* to ensure high-quality annotations. We then discuss the *unique attributes* that make our dataset valuable for causal VideoQA tasks.

### 3.1 Dataset Construction

**Video source.** To focus on *visual reasoning*, we selected the timeless *silent* cartoon series, "*Tom & Jerry*" (1940)[1], spanning over **6 seasons** and **161** *episodes*. Silent cartoons, which lack other modalities like dialogue, foster visual reasoning in children. Similarly, we hypothesize that using silent cartoons for VideoQA will enhance visual reasoning. The Tom & Jerry series is ideal for a causal reasoning dataset, offering *abundant segments with diverse cause-and-effect relationships*. Concurrent work [22] on causal *image* generation also leverages Tom & Jerry.

**Annotations.** Each dataset sample includes the following annotations: {*Question*, *Answer*, *Explanation*, *start frame*, *end frame*}. Details in the following.

● **Questions** are crafted to capture the *why* or reasoning behind the *actions* of characters in Tom & Jerry cartoons. To cultivate comprehensive & deeper video understanding capabilities in models, we formulated *thought-provoking* causal questions where cause & effect are connected by *longer causal chains*. Questions also extend *beyond explicit* visual cues, encompassing gestures & expressions, to delve into the *characters' underlying intentions & goals*. Annotators generated questions while watching the video for the first time to mimic how models assess unseen clips. They then rewatched the videos multiple times to create more critical thinking questions. To focus on visual reasoning, annotators watched the videos without audio, ensuring no audio cues were included in the dataset.

● **Multi-level Answers**. The first-level or the Primary answer, represents a *literal* or *direct* cause or form of response. It is accompanied by a deeper form of Explanation, which considers the *broader context* of the scene, the *thoughts*, *feelings*, *intentions* of characters, & their actions. This deeper explanation also takes into account potential consequences & provides further reasoning to support the primary answer. It includes reasons & additional details to comprehensively address the question. Examples shown in Figure 1 & Appendix. **Guiding principle for consistency:** our dataset's multi-level answer structure is designed to explore causality from direct causes (Primary answers) to deeper, contextual reasoning (Explanations). To ensure consistency, each Explanation must extend the causal chain from the Primary answer with at least one additional causal link, providing the next logical step in causal chain without unnecessary detail. For instance, if the Primary answer addresses an immediate reaction, the Explanation would delve into underlying motivations, intentions, or consequences, systematically adding depth. **Commonsense knowledge:** understanding causal relationships in complex scenes, such as those in Tom & Jerry episodes, often requires more than just visual cues—it requires the integration of commonsense knowledge. When commonsense knowledge is needed to connect cause-and-effect, annotators articulate this understanding within the Explanation.

● **Temporal annotations**. For each QAE set, the *Start* and *End frame numbers* are recorded to cover the entire scene, including contextual frames that support the reasons behind the actions.

**Quality Check.** To ensure dataset quality and reliability, we implemented a two-stage quality check process. First, quality checkers assessed episodes they did not annotate. Then, they reviewed episodes they did not check in the first stage. Multiple quality checkers carefully reviewed and verified curated question and answer sets for logical fallacies, timestamp inconsistencies, grammatical errors, and

---

[1]We only provide the annotations; Videos can be obtained from: https://www.warnerbros.com/tv/tom-and-jerry-1965-volume-1.

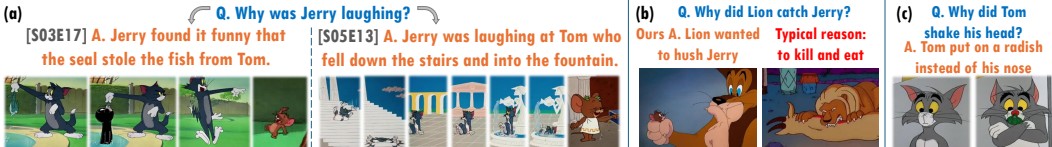

Figure 3: **Grounded in diverse visual information.**

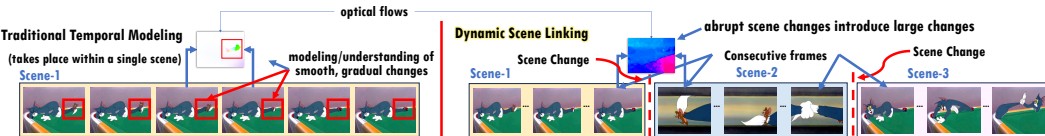

Figure 4: **Traditional temporal modeling vs Dynamic scene linking.** Notice the abrupt scene change, which causes disruption in visual flow, resulting in large amplitude and widespread optical flow.

spelling mistakes. They also excluded annotations containing audio or text aspects that the model cannot comprehend. To prevent direct overwriting, edits were flagged in a different font color for further discussion. Flagged inconsistencies or errors were resolved through discussions and consensus among checkers and annotators. The team referred to episode synopses from Tom and Jerry Wiki [45], an online encyclopedia, as a third-party opinion and for fact-checking.

**Dataset Statistics.** Annotators reviewed all **161** episodes across **6** seasons, generating **4,945** detailed Question-Answer-Explanation sets. This comprehensive dataset covers various scenes, characters, and events, providing a solid foundation for understanding and answering questions about the series. Additional statistics and word clouds for answers and explanations are available in the Appendix.

## 3.2 Unique Characteristics of our CausalChaos! Dataset

In the following, we delve into the untapped unique properties of our dataset, which both *complement existing video understanding datasets* and present *novel challenges to VideoQA*, particularly from a *causal reasoning perspective*.

1. **Multi-level answers** offer a *richer and more nuanced perspective*, which leads to *deeper insights*—considering not only the *immediate circumstances* but also underlying *psychological*, *social*, and *personal* factors that contributed to *characters' decisions*. By employing *multi-level answers alongside thought-provoking questions*, we can cultivate *deeper reasoning* and *analysis of characters' actions*, enabling the training and evaluation of models on complex question-answer pairs that demand a *comprehensive understanding* of *longer causal chains*. Further elaborated discussion on the importance of multi-level answers to 'Why'-questions included in Appendix.

2. **Grounded in diverse *Visual & Motion* information.** The Tom & Jerry cartoon series is *visually rich* and *dynamic*, with *intricate scenes*, *actions*, and *character interactions varying across episodes*. We had annotators craft questions where the *answer is grounded in the video*. This grounding compels the *model to analyze the video* for a *broader range of details* and *clues* to provide *meaningful* answers. For example, consider Figure 3(a), same question appears in different episodes, but their answers/context are completely different. This demonstrates that it is crucial for VideoQA models to understand visual information to answer correctly on our dataset. Multimodal nature of VideoQA task is enhanced by the presence of *unusual situations* in cartoons—typically not found in real-world datasets as shown in Figure 3(b,c). Since the answers are embedded in videos, models cannot rely on *correlations/biases* in their training data; they must thoroughly process and understand the content to answer correctly. This is crucial even with large language models (LLMs) and embeddings, which are also susceptible to *frequency-based biases* and hallucinations [20, 25, 62]. For example, *Lion* is often associated with *killing* in LLM embeddings. In a qualitative analysis, we found that models might choose an answer involving '*killing*' Jerry as the *motive* for the Lion-character's action, while the Lion-character was actually trying to *quiet* Jerry Figure 3(b-left). Revealing such biases has been considered in other computer vision problems [4, 12], but is yet to be introduced in VideoQA. *Our dataset offers a valuable opportunity to address this challenge.* Additionally, understanding Tom

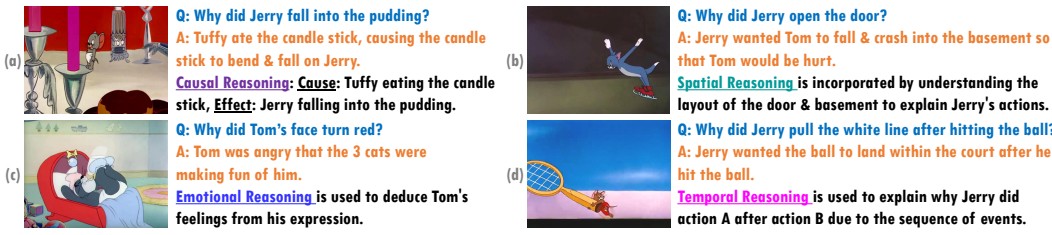

Figure 5: **Examples of various types of reasoning required by our dataset.** *Please zoom in & view in AdobeReader to play the embedded videos.* Further types of reasoning visualized in Appendix.

& Jerry cartoons requires modeling of *details* and *actions at varying levels of granularity*—from *sweeping movements* to *subtle emotional cues* via facial expressions.

3. **Focus on causal reasoning in visually dynamic scenes.** Our dataset focuses on causal reasoning in Tom and Jerry cartoons, characterized by dynamic and rapidly changing scenes. Characters and objects may appear, disappear, and reappear, limiting the model's access to partial observations. Video understanding models struggle to track and understand context. To answer causal questions, these models must link events to form a causal chain despite rapid scene changes and dynamic interactions. We term this Dynamic Scene Linking (DSL), distinct from traditional temporal modeling, which typically focuses on the gradual transitions within a scene as shown in Figure 7 (more details in Appendix). Related challenges are noted in other computer vision problems [36, 47] but not yet in VideoQA. We hypothesize that humans understand cartoons by forming a mental "world model" of the scene, which helps bridge gaps between discontinuous scenes and provides a coherent overall context. Current VideoQA datasets often lack coverage of such complex scenarios, unlike our dataset.

4. **More challenging and requiring cognitive effort.** We used GPT-4o [33] to compute the lengths of causal chains of *100 randomly chosen QAE pairs* from *our dataset* and *existing causal(-Why) VideoQA datasets* [49, 17, 19]. Results are presented in Figure 2. We observed that the QAE pairs in *our dataset have longer causal chains than existing QA datasets*, potentially suggesting our QAs are more complex. Identifying and solving *longer causal chains* in the process of answering causal (why) questions involve linking together multiple cues/clues, which requires significant cognitive efforts on the part of VideoQA models, especially, when the clues/cues are embedded in different scenes (described above), as in our dataset.

5. **Leveraging principles of animation** [44] (shown in Figure 1) such as *timing*, *squash and stretch*, *anticipation*, *staging*, and *exaggeration* can be conceptualized as **Data Design-driven Visual Prompting**, as they aid in: 1) "*highlighting*" *key movements*, *emotions*, & *storytelling*; 2) Consequently, greatly aid in *establishing clear cause-and-effect relationships* and *effectively communicating them*. These principles serve as *spatiotemporal* counterparts of *caricatures*, which exaggerate and manipulate facial and bodily features, and these have been shown to improve facial recognition rates [39, 11, 30]. In other domains of computer vision, some works [43, 52, 51, 5] acknowledge the advantages and devise methods to *disentangle the area of interest*, enabling more *targeted* processing and *comprehension of image contents*. Similarly, we hypothesize that the *principles of animation* can provide "*hints/guidance*", but the models still need to be able to *leverage this guidance to solve causal relationships*. Overall, *causal relationships in our dataset are complex, longer*, but at the same time they are *unambiguous/well-defined* using principles of animation, allowing models to *focus on deciphering causal relationships*.

6. **Wide spectrum of types of reasoning required.** Our dataset demands various types of reasoning like 1) *Deductive*; 2) *Inductive*; 3) *Spatial*; 4) *Causal*; 5) *Critical thinking*; 6) *Emotion*; 7) *Abductive*; and 8) *Temporal*. We have visualized examples demonstrating these reasoning types in Figure 9. It is also possible that a question may involve more than one type of reasoning. We conducted *human studies* to determine the types of reasoning required by our dataset and NextQA and *observed that our dataset demands a wider range of reasoning than NextQA* (results in Figure 2a). More information and definitions of types of reasoning and human studies are provided in Appendix.

## 3.3 Benchmark Design

**Question-types/Tasks.** Our dataset comprises two types of questions. The first is Multiple Choice Question Answering (**MCQA**). VideoQA models are provided with the question, the associated video

footage, **1** *correct* answer, and **4** *incorrect* answers; the task for the models is to pick the correct answer. The same format is followed for Answers and Explanations. The second is Open-ended Answer Generation (**OEAG**). The models are provided only with the question and the video footage, and the models have to generate the Answers and Explanations in natural language.

**Incorrect Answer/Option Mining.** In designing **MCQA** sets, answer options should be *distinct* from the correct answer & each other but *semantically similar enough to challenge reasoning beyond commonsense*. This prevents models from taking *shortcuts* like simple object/action matching in vision-language spaces. Using *randomly chosen answers* as incorrect options is *unproductive* and makes the correct answer *too obvious*, so we avoid this strategy. Instead, we introduce these strategies:
1. **Vanilla Hard Negative mining.** A pre-trained Sentence-Bert measures semantic similarity. For each question, we sample the top-10 other questions based on cosine similarity of Sentence-Bert embeddings. From these, we sample associated answers and again apply cosine similarity between the correct and candidate answers. Using the equation $\alpha cossim(Q_i, Q_n) \cdot \beta cossim(A_i, A_n)$, with adjustable weights $\alpha$ and $\beta$, we select the top 4 answer candidates. To avoid overly similar answers, we penalize high similarity scores by setting $\beta$ to 0.001, reducing the candidate's overall score.
2. **Causal-Confusion Negative generation. Causal relationship modeling** is *crucial* for causal VideoQA, but maybe given less emphasis in existing datasets/benchmarks. We hypothesize that, *relationships might not be adequately modeled* by existing models, instead models may be *selecting answers based on basic object/actor matching* in vision-language spaces. To test our hypothesis & underscore the relationship modeling, we introduce *hard negatives* in which the *objects/actors remain unchanged*, but the *relationships between them* are **altered** or alternate, but plausible scenarios are created. For instance, the *effect is inverted* (*e.g.*"Tom was hitting Jerry"→"Tom was *not* hitting Jerry") or the *causal agents are swapped* (*e.g.*"*Tom* was hitting *Jerry*"→"*Jerry* was hitting *Tom*"). We leverage LLMs to generate *causally-incorrect* options. Examples can be found in the Appendix.

**Design of Train-Val-Test splits.** We typically partition the dataset into **70%** for training, and **15%** each for validation and testing. Additionally, we employ specific strategies for training and testing.
1. **Uniformly Distributed *Seen*-episode testing (UD).** Dataset samples are randomly divided into train-val-test splits without constraints on which episodes or parts are included in each split. Consequently, uniformly distributed chunks of **all** episodes from **all** seasons are likely seen during training, potentially giving VideoQA models *sparse storylines*.
2. **Consecutive Partially *Seen*-episode testing (PS).** The *train* set is drawn from the **first 70%** of each episode, the *val* set from the **middle 70-85%**, & the *test* set from the **last 15%**. This setup allows the model to have partially "seen" episodes during training & make "educated guesses" during testing.
3. ***Unseen*-episode testing (UN).** In this case, a certain **70%** of the *episodes* are reserved for *train* set; **15%** of the remaining episodes are reserved for *val* set; and the *remaining* **15%** of the episodes are used to prepare the *test* set. So, *testing* is done on entirely novel, *unseen episodes* and *storylines*.

## 4 Experiments

We benchmark SOTA VideoQA methods on our new causal action QA dataset for two tasks: 1) **MCQA** and 2) **OEAG**. Then, we explore if our dataset aids real-world QA.

### 4.1 Benchmarking on MCQA

**Baselines.** Following prior work [49], we benchmark the performance of these models on our causal action QA dataset: *BlindQA* [2], *EVQA* [2], *CoMem* [8], *HME* [6], *HCRN* [16], and *HGA* [15]. We also evaluate recent vision-language models like *MIST* [7], *BLIP-2* [18], and multimodal instruction tuning models *Video-LLaMA* [59] and *VideoChat2* [21], *GPT-4o* [33], *VILA* [24] which excel on vision-language tasks. Further details on each model are provided in Appendix.

**Evaluation Protocols.** Models are evaluated under these protocols, with **Accuracy** as the metric:
• **Protocol 1.** Models' ability to select the *correct Answer (A)* is only evaluated. Their ability to select the correct explanation is not taken into consideration. The chance accuracy in this protocol is $1/5$. Results from this protocol are presented under columns marked as **A** in Table 1.
• **Protocol 2.** Models' ability to select the *correct Answer (A) as well as the correct Explanation (E)* is evaluated. If the model can select the correct answer, but not the correct explanation, then the

model is considered to have failed. In this protocol, the chance accuracy becomes $1/5 \times 1/5 = 1/25$. Results from this protocol are presented under columns marked as **A+E** in Table 1.

**Quantitative results.** The performances of the baseline models for both protocols are presented in Table 1. In general, we observe that the performances of most of the baseline approaches are low. Most of the models gain only around 11-12% improvement over the chance accuracy for both protocols. MIST seems to be doing exceptionally well compared to other models. We hypothesize that this could be because MIST being geared for long-form video understanding, is better able to handle long video contexts, which are frequent in our dataset. **Analysis**: we observed the following **failure modes** for models: **1) Limited Evidence Consideration**: Models often focus on a small subset of evidence, neglecting other relevant cues distributed throughout the sequence, which is problematic for datasets with long causal chains like ours. **2)** Models failed because they tried to exploit **shortcuts** like *object-noun* or *action-verb matching* in video-language spaces instead of focus-

| Model | UD | | PS | | UN | |
|---|---|---|---|---|---|---|
| | **A** | **A+E** | **A** | **A+E** | **A** | **A+E** |
| Chance | 20.00 | 04.00 | 20.00 | 04.00 | 20.00 | 04.00 |
| *Finetuned* | | | | | | |
| **BlindQA** | 29.38 | 13.07 | 26.51 | 11.54 | 25.13 | 11.02 |
| **EVQA** | 29.38 | 13.48 | 31.32 | 13.32 | 27.82 | 14.65 |
| **CoMem** | 32.08 | 13.88 | 26.10 | 09.89 | 23.12 | 09.27 |
| **HME** | 32.35 | 14.02 | 29.53 | 12.36 | 25.13 | 10.22 |
| **HCRN** | 32.48 | 16.98 | 32.01 | 14.97 | 25.67 | 12.23 |
| **HGA** | 31.40 | 15.36 | 29.40 | 13.19 | 28.23 | 13.84 |
| **MIST** | 62.22 | 44.88 | 62.22 | 42.86 | 55.91 | 37.90 |
| **MIST-CC**[†] | 63.34 | 46.80 | 62.50 | 43.54 | 56.18 | 39.25 |
| **VILA** | 77.22 | 62.80 | - | - | - | - |
| *Zero-shot* | | | | | | |
| **BLIP-2** | 43.67 | 23.32 | 45.88 | 24.18 | 46.64 | 26.48 |
| **Video-LLaMA** | 35.00 | 11.73 | 35.16 | 9.34 | 29.62 | 9.68 |
| **VideoChat2** | 38.14 | 15.36 | 38.91 | 15.93 | 40.83 | 15.99 |
| **GPT-4o** | 63.64 | 48.17 | 63.84 | 49.79 | 64.82 | 52.23 |
| **Humans** | 94.80 | 93.40 | - | - | - | - |

Table 1: **MCQA Results on our dataset.**[†]We design multitask version of MIST that learns to generate causal chains as an auxiliary task.

ing on causal relationship modeling—such *shortcuts can lead to wrong predictions on our dataset due to our hard negative mining*, correct & incorrect answers would likely contain these nouns/verbs. The most effective way to discriminate on our dataset is by inferring the causal relationship. We have discussed *further shortcomings* & **Qualitative Results** in Appendix. To mitigate these shortcomings, we design a MIST-based multitask model, **MIST-CC**, which, in addition to doing VideoQA, also learns to generate the causal chain from Video-Question features (groundtruth causal chains are generated by GPT-4 from QAE pairs). We found that this simple modification mildly boosts the performance. We have provided further details on MIST-CC in Appendix. Moving to VLM/MLLMs, we observe that they perform better than traditional models ([2, 15, 16]), except MIST. However, GPT-4o released only in late May 2024, outperformed all the models. Although, it is closed source model, we hypothesize its vision and language capabilities maybe significantly better than other VLMs. It was able to incorporate small details like facial expressions. To glean insights into it, we also asked it to give us its reasoning, and found that it analyzes each option individually and then selects the most likely answer. This is close to (or at least mimicking) how humans would approach this task. We believe this, at least on surface, seems to be going beyond just correlation-based answer picking as in non-VLM/MLLNs. In comparison, we found that **humans** performed significantly better than all the models. We established the human baseline on our dataset (MCQA) using five human subjects who are fluent in English and have at least an undergraduate level of education. This baseline was based on a subset of our dataset, consisting of 100 randomly chosen samples. We also provide results for the recent VLM, VILA [24] (CVPR 2024), *fine-tuned* and evaluated on our dataset. These results were obtained following our initial paper submission.

**Comparing the two protocols**, we find that Protocol 2, where the model has to select both the correct answer and the correct explanation is significantly more difficult than selecting just the correct answer for all the models. **Comparing splits**, We observe that UD is the easiest split across all models, followed by PS, with UN being the most difficult. This intuitive order suggests that understanding past events or storylines may aid in reasoning about current events or actions.

We further conducted an experiment where we tested the best performing models, MIST, MIST-CC, & GPT-4o, on our **Causal-Confusion** set, where incorrect answer choices have the same objects and actors as the correct answer option, but cause-and-effect relationships are reversed or altered. We observed a *significant drop* in performance of all models as shown

| Negative type | MIST | MIST-CC | GPT-4o |
|---|---|---|---|
| Vanilla Hard | 62.22 | 63.34 | 63.64 |
| Causal-Confusion | 55.80 | 58.76 | 54.95 |

Table 2: **Impact of causal-confusion.**

in Table 2. This could potentially be due to *causal relationships being not modeled adequately* even by such advanced VideoQA models. Performance of MIST-CC dropped relatively less, potentially because the auxiliary task of generating causal chains may have enhanced the understanding of causal relationships. Interestingly, we also found the S-BERT similarity score to be above 90% for an action

(*e.g.*, Tom is running after Jerry) and its Causal-Confusion version (*e.g.*, Tom is *not* running after Jerry); while these sentences would be opposite/different in terms of human perception. We believe that *Causality might be more overlooked than we think in various fields, not just in computer vision.*

## 4.2   Benchmarking on Open-ended Answer Generation (OEAG)

**CapsMIX Performance metric.**   We measure the performance of OEAG in terms of BLEU-1,2,3 [35], METEOR [3], ROUGE [23], SPICE [1], CIDEr [46] & Sentence-BERT [38] scores by comparing with the ground truth answers as done by captioning & QA literature. The wide range of metrics complicates model comparison, so we introduce Caps-MIX (Captioning Metrics Integration eXpert), which normalizes and integrates all scores into a single metric, simplifying comparisons and combining the unique strengths of individual metrics.

**Baselines.**   Following prior work [49], we benchmarked the performance of *EVQA* [2]; *UATT* [53]; *HME* [6]; *HGA* [15], *VILA* [24] on OEAG task. We also report the zeroshot performance of recent multimodal video understanding models *BLIP-2* [18], *Video-LLaMA* [59], *VideoChat2* [21], *GPT-4o* [33].

| Model | CapsMIX |
|---|---|
| *Finetuned* | |
| BlindQA | 2.7646 |
| UATT | 3.3928 |
| HME | 3.0975 |
| HGA | 3.7872 |
| BlindGPT-2 | 6.6006 |
| VisionGPT-2 | 6.7582 |
| VILA | **8.0320** |
| *Zero-shot* | |
| BLIP-2 | 1.8931 |
| Video-LLaMA | 2.4464 |
| VideoChat2 | **3.9524** |
| GPT-4o | 2.9851 |

Table 3: **OEAG results on our dataset (UD split).**

We evaluated the baselines for generating answers and explanations. The performances of various models are summarized in Table 3. Here we have presented unnormalized performances. For full results, and normalized version, see Appendix. Overall, we observe that models struggle significantly with open-ended generation, including some recent VLMs and MLLMs. Despite GPT-4o's strong performance on MCQA, it also falters on OEAG. These models likely perform better on MCQA by *eliminating incorrect choices*, but they fail to genuinely understand videos and perform causal reasoning for standalone answer generation. VideoChat2 seem to be doing well, perhaps, because it was specifically trained on various video understanding tasks and datasets, including causal reasoning task. Comparing split-type wise, we found that as in the case of MCQA, UD was the easiest split.

Inspired by the success of LLMs, we experimented with leveraging GPT-2 [37], a publicly available LLM, as our natural language answer generator. We found that a pre-trained version out of the box did not work well—it generated mostly random, unrelated words. However, upon simple training on the train set, it performed significantly better than the baselines we considered as shown in Table 3 (BlindGPT-2). However, it is unclear if these LLMs have "seen" Tom and Jerry scripts during their pretraining stage. If so, then pretraining on relevant scripts followed by finetuning on our dataset could potentially be a reason for GPT-2's good performance. Although this is less likely. What is more likely to be the reason behind this good performance is the language modeling/generating capability of GPT-2/LLMs.

In the next step, we integrated visual information into LLM. Taking inspiration from [32], we learn a projector network to align the visual features with GPT-2 representations (VisualGPT-2 in Table 3. We observed an improvement in the performance. However, we believe that this might not be a very efficient way to integrate visual information with LLMs. We expect the performance to boost considerably through better and more sophisticated joint modeling vision and language. Overall, we believe shallower networks might not have the capacity to do inference over longer, complex causal chains, and as such it might not be the best option to invest future efforts into; LLMs have the potential to excel at causal reasoning; this view is also supported in a concurrent survey [26]. We have also provided results of VILA [24] finetuned on our dataset. These results were obtained following our initial paper submission. Note that none of the models discussed are performing well. This becomes clearer when we compare their performances with the upper bound (see the normalized version in Table 6). This shows that there is a significant room for improvement on OEAG task.

## 4.3   Does our dataset help with real-world cases?

To evaluate the direct impact of our dataset, we combine our training set with NextQA's [49] training set and measure model performance on the NextQA [49] test set. For comparison, we also measure model performance without incorporating our dataset. Despite being a *synthetic/cartoon* dataset, and

*much smaller in size*, our dataset boosted the performance on a real-world dataset on both MCQA and OEAG (Table 4).

We observed *improvement in identifying the correct causes by breaking the reliance on shortcuts & focus on causal effects*; a more comprehensive analysis of the situation/interaction, rather than jumping to a conclusion. Qualitative results provided in Appendix. What is more, improvements were not limited to 'causal-why' questions but also extended across other question types. We hypothesize that this can attributed to the *wider range of reasoning* involved in our dataset. To the best of our knowledge, this is the first time a synthetic VideoQA dataset has shown immediate improvement on a real-world dataset.

| Train Data | MCQA | | | | OEAG | |
|---|---|---|---|---|---|---|
| | BlindQA | HGA | MIST | VILA | HGA | VILA |
| NextQA | 32.73 | 41.23 | 55.44 | 67.23 | 1.44 | 2.66 |
| + Ours | **32.91** | **41.44** | **55.96** | **68.46** | **1.48** | **2.86** |

Table 4: **Our dataset improves performance on existing real-world dataset.**

Although modest, the improvements are meaningful because: **1)** improvements held across models showing broad applicability; **2)** improvements beyond why-questions suggest that the models' enhanced causal reasoning skills and learnt representations generalize beyond the specific questions seen during training, potentially improving overall video comprehension; **3)** Cross-domain learning often carries a high risk of negative transfer, where irrelevant or misleading patterns from one domain can harm performance in another. The fact that the cartoon dataset did not degrade performance across multiple models is a good indication that it introduces valuable, complementary information.

However, we noticed a slight drop in the performance on location-type questions. Thus, we believe that it might be better to: **1)** *transfer reasoning skills acquired from challenging synthetic datasets like ours*; **2)** leverage our synthetic dataset to *inform the model designing process*, as it better reflects the challenges VideoQA models may face in the real world, such as longer causal chains and frequent scene changes in visual streams. Nonetheless, we *do not suggest naively deploying whole models/weights trained on our dataset to real-world scenarios or applications*.

## 5 Conclusion

We introduced CausalChaos!, a challenging dataset for causal action question-answering tasks based on the classic Tom and Jerry cartoon, richly annotated with critical thinking questions requiring extensive reasoning from Video QA models. Questions come with multi-level answers and explanations covering the entire video context. We also provide the novel CausalConfusion test set to challenge causal relationship modeling in Video QA models. Our experiments show that while existing models perform well on causal action QA tasks, there is significant room for improvement in causal relationship modeling and generating detailed open-ended answers. LLMs show promise, but integrating visual information with LLMs or joint modeling of vision and language is crucial. We hope our dataset fosters such developments and will release it and the codes to support future efforts. Lastly, we demonstrated improvements in real-world datasets. Our dataset, derived from cartoons, should inform model design, reflecting real-world challenges like longer causal chains and frequent scene changes. However, we do not suggest deploying models trained solely on our dataset in real-world scenarios.

**Material Acknowledgement and Disclaimer.** Tom and Jerry is a material of Turner Entertainment Company (Warner Bros. Entertainment Inc.). All rights reserved. We do not claim any ownership of or rights to the Tom and Jerry material. All other trademarks, service marks, trade names and any other material referenced in this document are the property of their respective owners.

**Acknowledgements.** This research/project is supported by the National Research Foundation, Singapore, under its NRF Fellowship (Award# NRF-NRFF14-2022-0001). This research is also supported by funding allocation to B.F. by the Agency for Science, Technology and Research (A*STAR) under its SERC Central Research Fund (CRF), as well as its Centre for Frontier AI Research (CFAR). We sincerely thank all the reviewers for their valuable feedback and suggestions.

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

# A   Appendix / supplemental material

We have provided the Appendix (**a PDF file + PowerPoint presentation**).

**Contents**

## A.1   Extended Related Work

### A.1.1   Datasets/Benchmarks

To drive the progress in Video QA, researchers have developed various datasets with distinct focuses and contributions. In the following, we will delve into the landscape of Video QA datasets, examining their characteristics, limitations, and the specific areas they address.

*Early Video QA Datasets.* During the early stages of Video QA research, several datasets relied on video captions or descriptions to automatically generate questions and corresponding answers. Examples of these datasets include *MovieFIB* [27], *MSVD-QA* [50], *MSRVTT-QA* [50], *YouTube2Text* [10], *open-ended QA*, *Zeng et al.*, and *Video Context-QA*[63]. These datasets played a crucial role in the initial exploration of Video QA but were primarily limited to object and action recognition. They lacked the ability to go beyond these basic visual cues, which posed limitations in understanding complex interactions and causal relationships within videos.

*TGIF-QA* [13, 14] focuses on short videos and relies on captions to generate questions and answers, but it is limited in its coverage of object interactions and causal reasoning. On the other hand, *ActivityNet-QA* [56] annotates longer web videos, offering a broader range of content, but it also lacks in capturing complex reasoning. Both datasets fall short in capturing the depth and complexity required for comprehensive video question answering; they did not fully explore questions and answers involving object interactions and causal relationships.

The *Social-IQ* [57] dataset is designed to address questions related to human social behavior in videos, relying on multimodal cues for answering. This dataset emphasizes the importance of understanding social interactions and dynamics within video content. By focusing on human behavior, Social-IQ offers a unique perspective in video question answering. However, it should be noted that this dataset primarily relies on multimodal cues, meaning that the answers to the questions heavily depend on

the combination of visual and other sensory information, and as such cannot be used as a visual reasoning dataset. While it provides valuable insights into social aspects of video content, the dataset may not fully capture the broader context and reasoning required for comprehensive video question answering.

*CLEVRER* [54] dataset covers temporal and causal relationships using collision events between various objects. Being limited to simple, inanimate objects and collision events, it does not cover reasoning involving emotions and intentions; objects do not have any characteristics; actions do not have motivation or rationale; limited set of events; scene does not have an actually involved background; character-object interaction is lacking. Moreover, the reasoning is over a shorter temporal horizon than ours. Since it is a synthetic dataset with programmatically generated QA pairs, there is also a lack of diversity in natural language descriptions of the events and human judgments. *CLEVRER-Humans* [29] bridges this language gap, but, other shortcomings still persist.

*AGQA* [9] focuses on spatio-temporal scene understanding. For example, Did they <action1> or <action2>for longer? What did the person do after <action>? What were they <action> first/last? *STAR*, a situation reasoning dataset, additionally, covers prediction and feasibility questions. However, they do not cover explanatory "why" questions like ours.

We have discussed and compared with *NextQA* [49], *CausalVidQA* [17], *IntentQA* [19] in detail in the main paper. Here we provide some additional details on them. NextQA [49] contains descriptive (related to location, counting, yes/no), temporal (related to temporal ordering previous, next), and causal questions. CausalVidQA [17] contains descriptive, causal, predictive, and counterfactual. While they provide a rationale for predictive and counterfactual, they do not explore and provide multi-level answers and explanations for Why-questions, while our dataset does provide them. IntentQA [19], a concurrent dataset explores understanding motivations based on context. Their dataset is derived from NextQA causal and temporal questions, but they construct their dataset in a contrastive manner such that the same actions under different contexts lead to different underlying intents.

### A.1.2 Models

In the following, we have discussed the state-of-the-art VideoQA models that we have benchmarked on VisCAQA dataset in the main paper. We have discussed their central concepts and unique design characteristics.

- *BlindQA* [2]. In this approach, no visual information is leveraged. Answers are chosen directly based on the questions. In a nutshell, this model learns a mapping from question to answer. Higher performance by method would suggest that the dataset contains questions that are not visually-grounded.

- *EVQA* [2]. This method extends BlindQA baseline by incorporating the visual stream modeled by an LSTM.

- *Spatio-Temporal Reasoning in Visual Question Answering (STVQA)* [13]. This work introduces three novel video QA tasks that demand spatio-temporal reasoning skills to answer questions accurately. In addition, a new TGIF-QA dataset has been created to facilitate research in this field. To address this issue, a dual-LSTM-based approach with both spatial and temporal attention mechanisms has been proposed as a baseline model.

- *Motion-Appearance Co-Memory Networks (CoMem)* [8]. A novel Video QA framework, combining Dynamic Memory Network (DMN) principles with motion and appearance features. This innovative approach leverages a co-memory attention mechanism to incorporate both motion and appearance cues. It employs a temporal conv-deconv network to create multi-level contextual information and utilizes a dynamic fact ensemble method for constructing dynamic temporal representations tailored to specific questions.

- *Heterogeneous Memory Enhanced Multimodal Attention Model (HME)* [6].This innovative end-to-end trainable Video QA framework begins by generating global context-aware visual and textual features. It achieves this by interacting the current inputs with memory contents. Subsequently, it integrates these multimodal features through attentional fusion to make accurate inferences for answering questions.

- *HCRN* [16]. This is a hierarchical framework with conditional relation networks as building blocks models input video at multiple scales (clip-, full video-level) in a cascaded manner.

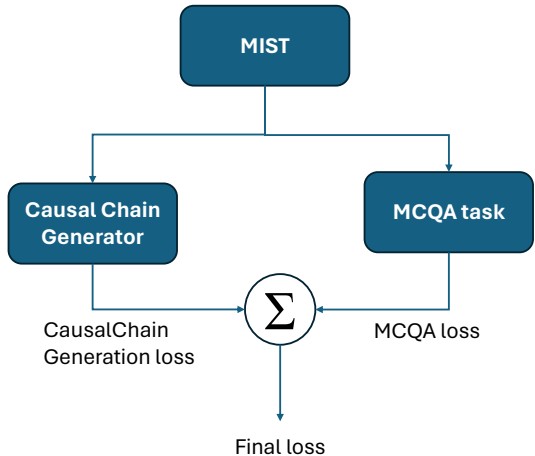

Figure 6: **MIST-CC framework.**

Visual features at each level are conditioned on the question features. The joint representation is fed into the classifier for answer prediction.

- *HGA* [15]. Leverages heterogeneous graph reasoning module and a co-attention unit to capture the local and global correlations between video clips, linguistic concepts and their cross-modal correspondences.

- *Multimodal Iterative Spatial-temporal Transformer (MIST)* [7]. MIST, designed for long-form Video Question Answering (VideoQA), revolutionizes conventional dense spatial-temporal self-attention. It accomplishes this by utilizing two critical modules: segment and region selection, which adaptively pick out frames and image regions tied to the questions. Following this, it processes diverse visual concepts effectively with an attention mechanism. This process occurs iteratively across multiple layers, empowering the model with multi-event reasoning capabilities.

### A.2 Implementation details

We use the publicly available github code repository: `https://github.com/doc-doc/NExT-QA` for BlindQA, EVQA, CoMem, HME, HCRN, and HGA.

### A.3 Details regarding MIST-CC

We design MIST-CC, a multitask version of MIST that learns to generate causal chains as an auxiliary task. With MIST-CC, our goal is not to generate perfect causal chains during testing, per se; but to focus on providing guiding signal to the model to improve the performance on question-answering task. Framework for MIST-CC is shown in Figure 6. We build upon the publicly available implementation of MIST. We implement the Causal Chain Generator in our MIST-CC framework using a single-layer gated recurrent unit (GRU) with a 1024-dimensional hidden state; and dropout rate of 0.2. The overall multitask (MTL) objective function to be minimized is the summation of: 1) multichoice question answering loss ($\mathcal{L}_{MCQA}$); 2) causal chain generation loss ($\mathcal{L}_{CCG}$) (Equation 1). We set $\alpha$ to 1. To obtain vanilla MIST, we set $\beta$ to 0; while to obtain MIST-CC, we set $\beta$ to 0.1. We use ADAM optimizer with an initial learning rate of 1e-4. We do not use learning rate schedulers. All the models are trained for 30 epochs, and the best version of a model is selected based on the performance on the validation set.

$$\mathcal{L}_{MTL} = \alpha\mathcal{L}_{MCQA} + \beta\mathcal{L}_{CCG} \tag{1}$$

### A.4 Annotators' Background

Five undergraduate students from computer science and electrical engineering disciplines were recruited as annotators. All the annotators listed at least Fluency as the English language skills level.

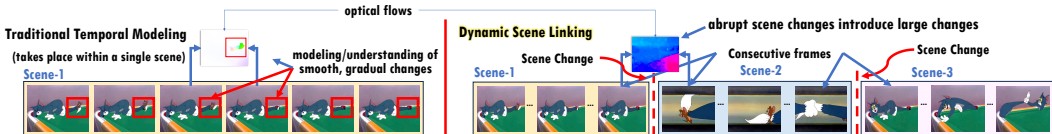

Figure 7: **Challenge of dynamic scene linking.** Here we have shown temporal modeling or understanding in a traditional sense; and compared it with dynamic scene linking. Our full causal chain for this particular example consisted of three scenes. In traditional temporal modeling, typically the dependencies are modeled over gradual transitions—typically, within a single scene. Notice the optical flow from traditional temporal modeling where Jerry is going into a pool table hole. On the other hand, notice the abrupt scene change, which causes disruption in visual flow, resulting in large amplitude and widespread optical flow. We have used optical flow as one way to illustrate the magnitude of change (but other measures may also be used).

## A.5 Dynamic Scene Linking

Our CausalChaos dataset involves more abrupt/frequent scene (event) changes than existing causal video QA datasets. Causal links or clues needed to solve causal relationships in QA pairs in our dataset are embedded in different scenes/events. Thus, video QA models must link these scenes/events together to understand the story. We term this problem Dynamic Scene Linking. In the following, we briefly discuss a related problem of temporal modelling and differentiate Dynamic Scene Linking from it. Although humans can seamlessly link such scenes, dynamic scene linking introduces a novel challenge for video understanding models in addition to temporal modelling.

Temporal modelling or understanding in video understanding generally refers to the process of analyzing and interpreting the temporal dynamics or changes within a sequence of frames in a video—typically within a single scene (refer to Figure 7). This involves capturing and understanding the patterns of motion, action, and context over time. Temporal modeling techniques aim to extract meaningful information from the temporal dimension of video data.

In the traditional sense, temporal modelling involves techniques that capture the gradual changes and transitions occurring within a video sequence. This includes methods like optical flow and 3D convolutional neural networks. These techniques are designed to capture the temporal dependencies and patterns of continuity or gradual evolution in videos.

On the other hand, abrupt scenes or shot changes, such as those found in cartoons like Tom and Jerry, represent sudden and significant shifts in the content or context of a video, however, these changes are causally linked. These changes can include shifts in location, characters, actions, or camera perspectives. Unlike gradual temporal changes, abrupt changes occur rapidly and may disrupt the continuity of the narrative or visual flow. While temporal understanding typically involves linking very nearby dependencies, dynamic scene linking involves linking across abrupt scene changes. For example, in Figure 7, in Scene-2, some of the things the model needs to be able to understand are: 1) the thing that Jerry is carrying is Tom's tail from its partial observation; 2) white furry hand is of Tom. Implicit causal reasoning plays a crucial role in establishing continuity between scenes, even when objects are seen partially or there are view changes. By relying on their understanding of cause-and-effect relationships within the narrative, humans can seamlessly integrate partial views and view changes into their mental model of the story. Similarly, models are required to do implicit causal reasoning for dynamic scene linking, and can benefit from incorporating capabilities such as forming a mental 'world model' of the story.

Temporal modeling techniques in the context of abrupt scene changes need to be able to detect and handle these sudden transitions effectively. While some traditional temporal modeling methods may capture gradual changes well, they might struggle to handle abrupt changes efficiently. Specialized algorithms or models may be required to identify and adapt to such abrupt scene changes.

## A.6 Full-size Tables and Figures

**Examples from CausalChaos! dataset.** For easier viewing, we have provided dataset video examples from Figure 1 from the main paper in the accompanying PowerPoint presentation.

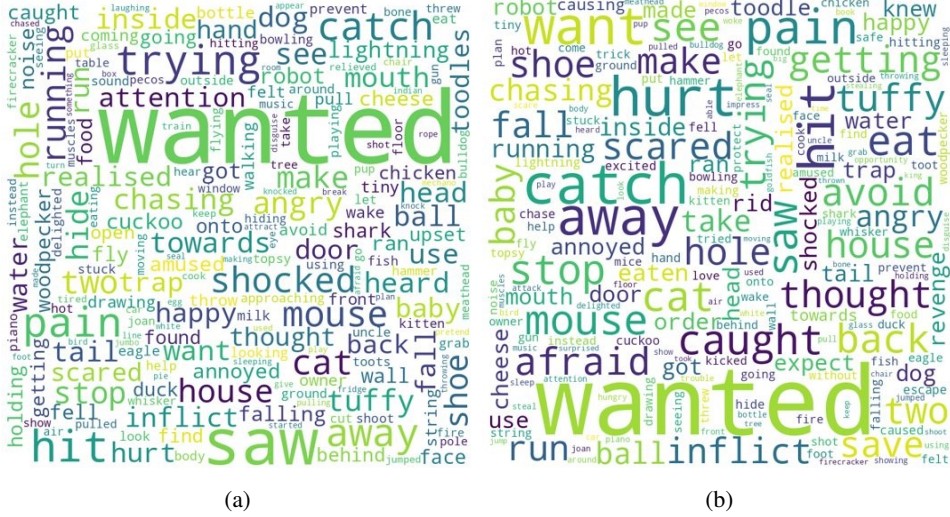

|       |       |
|:-----:|:-----:|
| (a)   | (b)   |

Figure 8: **Wordclouds** corresponding to (a) **Answers**; (b) **Explanations**.

**CausalChaos! vs NextQA dataset.** For easier viewing, we have provided dataset video examples from Figure 2(c) from the main paper in the accompanying PowerPoint presentation.

### A.7 Further Dataset Stats and Examples

**Dataset Stats:**

- Average clip length : 357.95 frames
- Longest of clips : 2315.0 frames
- Average length of question : 6.54 words
- Longest length question : 17 words
- Longest question : Why did Jerry and Tuffy put the wire into the water and turn on the freeze mode?
- Average length of Answer : 7.76 words
- Longest length Answer : 26 words
- Longest Answer : Tom saw his tail and hind legs at the top of the pipe while Tom's head and front legs were at the bottom of the pipe.
- Average length of Explanation: 13.88 words
- Longest length Explanation: 30 words
- Longest Explanation: Tom thought Jerry would walk into the hole and into Tom's mouth but Jerry let the toy mouse go first and Tom ate the toy mouse thinking it was Jerry.

**Dataset examples.** For easier viewing of the videos, we have provided them in the accompanying PowerPoint presentation.

### A.8 Wordclouds

Wordclouds are illustrated in Figure 8.

### A.9 Value in Multi-level Answers to Why-Questions

We richly annotate our dataset with multi-level answers to "Why"-questions behind the actions of characters in the Tom and Jerry cartoon. In this section, we discuss why and how a significant value lies in providing multi-level answers to "why" questions (or exploring various layers of causality or

explanation) regarding characters' actions (or, in general life, people's actions). "Why" questions do not always have simple, straightforward answers—they often involve multiple layers of explanation and understanding. *In our dataset, we consider cartoon characters and their actions, but here for more generality, we take human behavior as a case for discussion.* Human behavior is typically multifaceted, influenced by a variety of factors including personal experiences, emotions, social context, cultural background, cognitive processes, etc.

There can be multiple layers of rationale behind any action or decision. For example, someone might choose to volunteer at a homeless shelter. On the surface, the reason may seem obvious – to help those in need. But delving deeper, you might find additional motivations such as personal fulfillment, a desire to contribute to the community, religious beliefs, or even social pressure from peers.

Recognizing the complexity of human behavior and understanding that there can be multiple, intertwined reasons for why people act the way they do is essential for empathy, effective communication, and building strong interpersonal relationships. Some of the ways multi-level answers to 'Why'-questions help are as follows:

- Understanding Motivations: At the surface level, a person's actions may seem straightforward, but delving deeper can reveal the underlying motivations and intentions driving those actions. Multi-level answers can help uncover these motivations, providing a more comprehensive understanding of human behavior.

- Contextual Understanding: Human behavior is complex and influenced by a variety of factors, including personal experiences, societal norms, cultural background, and psychological factors. Providing multi-level answers allows for a more nuanced understanding of the context in which the behavior occurs, shedding light on the various influences at play.

- Predictive Insights: By understanding the multiple layers behind people's actions, it becomes easier to predict future behavior. Recognizing patterns in motivations and behaviors can help anticipate how individuals might act in different situations, enabling better decision-making and planning.

- Empathy and Compassion: Exploring the deeper reasons behind someone's actions fosters empathy and compassion. It allows us to see beyond the surface behavior and understand the person's perspective, experiences, and struggles, leading to more meaningful interactions and relationships. While this might be more of a human experience-related factor and might be limited to very specialized machines like empathetic robots, but still shows potentially how multi-level answers help.

- Problem Solving and Conflict Resolution: In situations where conflicts arise or problems need to be addressed, understanding the multi-level reasons behind people's actions can facilitate more effective problem-solving and conflict-resolution strategies. It enables individuals to address underlying issues rather than just surface-level symptoms.

Overall, providing multi-level answers to "why" questions behind people's actions enhances our understanding of human behavior, promotes empathy and compassion, and facilitates better decision-making and problem-solving.

## A.10 CausalChain Details and Examples

In the following, we have provided some examples from our dataset to illustrate causal chains of different lengths. Note that, answer and explanation were combined when computing the length of causal chains.

> Question: Why did Jerry slide?
> Answer+Explanation: Jerry slid down the clock because he saw Spike trying to catch Tom and was confident that Spike's attention was on Tom and not Jerry.

*Length of causal chain*: 2. In the given event involving Jerry, Spike, Tom, and a clock, we can identify several causal relationships that form a chain of events. Let's break it down:

1. Event A: Jerry sees Spike trying to catch Tom.
2. Event B: Jerry is confident that Spike's attention is on Tom.

3. Event C: Jerry slides down the clock.

Now, let's consider the causal relationships:

- Event A causes Jerry's perception of Spike's actions.
- Event B is influenced by Jerry's perception of Spike's actions.
- Event B causes Jerry's confidence in Spike's attention being on Tom.
- Event C is influenced by Jerry's confidence in Spike's attention.

So, we can identify at least two causal links or events in this scenario. Each event contributes to the next in a causal chain, leading to the final action of Jerry sliding down the clock.

> Question: Why did Jerry go below chicken?
> Answer+Explanation: Jerry went below the chicken sitting on her nest to hide and get protection from Tom.

Breakdown of the events and causal links in this case is:

- Event A: Tom poses a threat to Jerry.
- Event B: Jerry seeks protection and safety.
- Event C: Jerry goes below the chicken sitting on her nest as a protective measure.

Here, we can identify that there are *two* causal links in the causal chain of the event.

### A.11 Further details on Causal Chain Length Comparison Experiment

We leveraged GPT-4o [33] to compute the lengths of causal chains involved in the QA pairs from our and existing causal video QA datasets [49, 17, 19]. We randomly sampled 100 causal-Why QA pairs from all the datasets. Then, we used the following prompt: "What is the causal chain in the following question-answer pair? Please return the causal chain in the form of event_A->event_B->event_C...If no cause-and-effect relationship is addressed, then output 0...Question: [question added here] Answer: [answer added here]." to ask GPT-4o to obtain the causal chain in each QA pair. We define the length of a causal chain as the number of links in that chain. For example, "event_A->event_B" has a length of 1, while "event_A->event_B->event_C" has a length of 2. Since our dataset contains multi-level answers, we combined answers and explanations using GPT-4 to get an overall answer to reflect the true length of the full causal chain involved. We use these overall answers when computing the causal chains for our dataset. Extracted causal chains from all datasets were manually verified. Human verifiers agreed 89% of the times with the causal chains. Once the lengths of causal chains for all the samples are computed, we average them to get the average causal chain length for a dataset. We repeat the process for all datasets and then compare them.

### A.12 Details on Types of Reasoning

### A.12.1 Definitions of Types of Reasoning

In the following, we have provided the definitions of various types of reasoning.

1. **Deductive Reasoning** involves drawing specific conclusions based on general principles or premises. Questions from our dataset can require video understanding models to answer based on established patterns or cause-and-effect relationships between characters' actions (*e.g.*, Figure 9(a)).

2. **Inductive Reasoning** involves making generalizations or forming hypotheses based on specific observations. Tom and Jerry episodes contain such episode-specific actions or features or nuances, *e.g.*, as in Figure 9(b). Answering causal questions related to such actions involves inductive reasoning.

3. **Spatial Reasoning** involves predicting and understanding spatial relationships or configurations. Our dataset requires Video-QA models to have an understanding of the physical space

Figure 9: **Examples of various types of reasoning required by our dataset.** *Please zoom in & view in AdobeReader to play the embedded videos.*

and how the characters navigate it, interactions with the environment, including concepts such as distance, direction, and obstacles (crashing in or avoiding it). For example, as shown in Figure 9(c).

4. **Causal Reasoning** involves understanding cause-and-effect relationships between actions and their consequences. In the process of answering questions in our dataset, Video QA models will be required to engage in causal reasoning by linking the characters' actions or sub-events within an episode to the resulting consequences and understanding the cause-and-effect chains in the cartoon as depicted in Figure 9(d).

5. **Critical Thinking** in our setting encompasses a range of cognitive processes, including analysis, and evaluation by analyzing the visual cues, and interpreting the characters' actions. A way to judge the complexity of critical thinking questions is by measuring the lengths of causal chains. To get the full picture, it is also important how difficult each link in this causal chain is to be inferred from the video (refer to Figure 9(e))

6. **Emotion Reasoning** involves recognition and understanding of emotions and how they can affect behavior and decision-making. Our dataset requires models to perform emotion/facial expression recognition and link them to characters' actions/behaviors. For example, as shown in Figure 9(f).

7. **Abductive reasoning** involves making an inference or hypothesis based on limited or incomplete information, in order to explain or interpret a situation or phenomenon. Our dataset contains questions that involve making inferences from partial information, *e.g.*, it is to be inferred that Tom was scared because there is a fight going on from the visual cues of furniture being thrown around, without seeing the actual fight as shown in Figure 9(f).

8. **Temporal Reasoning** refers to understanding and reasoning about the sequence/ordering of events over time—understanding the relationships between different actions, and identifying causal relationships amongst them (Why A is done before/after B) as in Figure 9(g).

### A.12.2 Human Study Details

Human subjects in human studies had a background in the disciplines of computer science and electrical engineering; from undergraduate student level to postdoctoral level. Five human subjects participated in the study. For determining reasoning types, the subjects were first explained reasoning types and given brief training on identifying those. A screenshot is shown in Figure 10. Subjects were then shown the question-answer pairs (unseen during the briefing) and asked to choose the reasoning

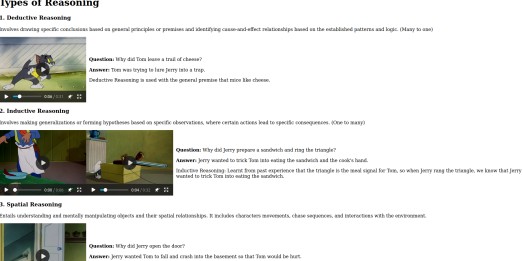

Figure 10: **Screenshot of a guide used as a part to explain types of reasoning to human subjects.**

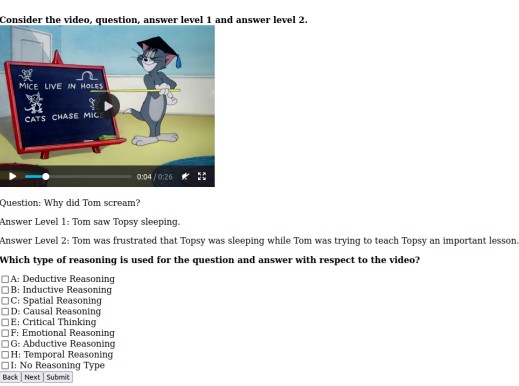

Figure 11: **Screenshot of user interface used for collecting responses from human subjects for CausalChaos! dataset.**

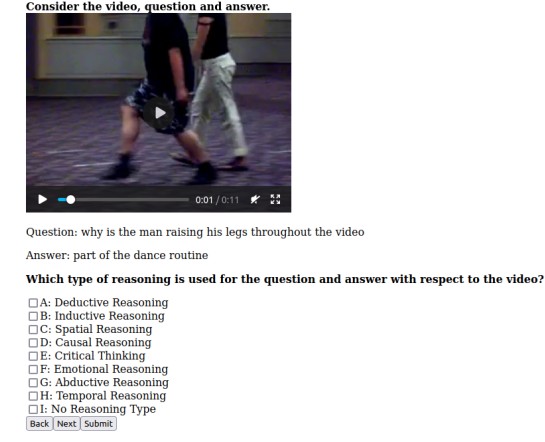

Figure 12: **Screenshot of user interface used for collecting responses from human subjects for NextQA dataset.**

types. Screenshots are shown in Figure 11, Figure 12. In the interface, all the reasoning types were listed out, and the subjects had the freedom to select multiple reasoning types if they thought an instance contained more than one type of reasoning. Additionally, a "No Reasoning Type" option was available to subjects in case they deemed that no reasoning was involved.

## A.13  Extended analysis of models' performance

In the main paper, we discussed two of the major limitations of VideoQA models. Other limitations include: 1) some models like MIST do not leverage explicit motion information using, *e.g.*, spatiotemporal convolutional neural networks. Due to this, they might inaccurately infer the scene based on a

| Model | BLEU-1 | | BLEU-2 | | BLEU-3 | | METEOR | | ROUGE | | SPICE | | CIDEr | | S-BERT | | CapsMIX |
|---|---|---|---|---|---|---|---|---|---|---|---|---|---|---|---|---|---|
| | A | E | A | E | A | E | A | E | A | E | A | E | A | E | A | E | |
| | | | | | | | | *Finetuned* | | | | | | | | | | |
| BlindQA [2] | 0.2412 | 0.1912 | 0.1039 | 0.0618 | 0.0415 | 0.0221 | 0.1243 | 0.0891 | 0.2732 | 0.1885 | 0.2956 | 0.0703 | 0.1034 | 0.0403 | 0.4987 | 0.4195 | 2.7646 |
| UATT [53] | 0.2661 | 0.2203 | 0.1257 | 0.0805 | 0.0575 | 0.0330 | 0.1409 | 0.1046 | 0.3171 | 0.2293 | 0.3547 | 0.2775 | 0.1197 | 0.0397 | 0.5495 | 0.4767 | 3.3928 |
| HME [6] | 0.2650 | 0.2072 | 0.1301 | 0.0713 | 0.0554 | 0.0219 | 0.1504 | 0.0914 | 0.3125 | 0.2273 | 0.3184 | 0.0826 | 0.0936 | 0.0639 | 0.5328 | 0.4737 | 3.0975 |
| HGA [15] | 0.2891 | 0.2263 | 0.1588 | 0.0897 | 0.0789 | 0.0323 | 0.1856 | 0.1130 | 0.3388 | 0.2399 | 0.3980 | 0.2426 | 0.2782 | 0.0573 | 0.6026 | 0.4561 | 3.7872 |
| BlindGPT-2 | 0.3770 | 0.3165 | 0.2530 | 0.1749 | 0.1632 | 0.1091 | 0.2452 | 0.1761 | 0.3858 | 0.2966 | 0.4305 | 0.3529 | 1.2482 | 0.7690 | 0.6755 | 0.6271 | 6.6006 |
| VisionGPT-2 | 0.3878 | 0.3095 | 0.2605 | 0.1727 | 0.1738 | 0.1091 | 0.2560 | 0.1756 | 0.3934 | 0.2941 | 0.4498 | 0.3385 | 1.3725 | 0.7539 | 0.6760 | 0.6350 | 6.7582 |
| VILA (FT) [24] | 0.3941 | 0.3333 | 0.2907 | 0.2111 | 0.2088 | 0.1391 | 0.2972 | 0.2221 | 0.4417 | 0.3532 | 0.4930 | 0.4342 | 1.7110 | 1.0427 | 0.7331 | 0.7264 | **8.0320** |
| | | | | | | | | *Zeroshot* | | | | | | | | | | |
| BLIP-2 [18] | 0.1381 | 0.0815 | 0.0451 | 0.0256 | 0.0167 | 0.0059 | 0.0664 | 0.0480 | 0.1618 | 0.1312 | 0.0530 | 0.0422 | 0.2279 | 0.1046 | 0.3837 | 0.3614 | 1.8931 |
| Video-LLaMA [59] | 0.1241 | 0.1181 | 0.0419 | 0.0344 | 0.0115 | 0.0098 | 0.1477 | 0.1163 | 0.1719 | 0.1435 | 0.2055 | 0.1383 | 0.0836 | 0.0430 | 0.5734 | 0.4834 | 2.4464 |
| VideoChat2 [21] | 0.2353 | 0.2116 | 0.0823 | 0.0776 | 0.0250 | 0.0253 | 0.1769 | 0.1295 | 0.2667 | 0.2168 | 0.3264 | 0.2547 | 0.3980 | 0.2910 | 0.6445 | 0.5908 | **3.9524** |
| GPT-4o [33] | 0.1449 | 0.0814 | 0.0522 | 0.0253 | 0.0171 | 0.0060 | 0.1975 | 0.1530 | 0.1670 | 0.1064 | 0.3169 | 0.1808 | 0.2601 | 0.0029 | 0.6697 | 0.6039 | 2.9851 |

**(a)**

| Model | BLEU-1 | | BLEU-2 | | BLEU-3 | | METEOR | | ROUGE | | SPICE | | CIDEr | | S-BERT | | CapsMIX |
|---|---|---|---|---|---|---|---|---|---|---|---|---|---|---|---|---|---|
| | A | E | A | E | A | E | A | E | A | E | A | E | A | E | A | E | |
| | | | | | | | | *Finetuned* | | | | | | | | | | |
| BlindQA [2] | 0.2193 | 0.1795 | 0.0772 | 0.0468 | 0.0326 | 0.0149 | 0.1122 | 0.0813 | 0.2501 | 0.1767 | 0.2960 | 0.1147 | 0.1099 | 0.0434 | 0.5129 | 0.4260 | 2.6935 |
| UATT [53] | 0.2693 | 0.1947 | 0.1257 | 0.0581 | 0.0440 | 0.0146 | 0.1466 | 0.0847 | 0.3226 | 0.2107 | 0.3255 | 0.2012 | 0.1444 | 0.0646 | 0.5364 | 0.4829 | **3.2260** |
| HME [6] | 0.2475 | 0.1830 | 0.1031 | 0.0549 | 0.0363 | 0.0154 | 0.1417 | 0.0845 | 0.2820 | 0.2173 | 0.2933 | 0.2549 | 0.0831 | 0.0655 | 0.5165 | 0.4929 | 3.0719 |
| HGA [15] | 0.2586 | 0.1842 | 0.1085 | 0.0517 | 0.0365 | 0.0148 | 0.1433 | 0.0932 | 0.2909 | 0.1806 | 0.2908 | 0.2128 | 0.0877 | 0.0266 | 0.5231 | 0.4078 | 2.9111 |

**(b)**

| Model | BLEU-1 | | BLEU-2 | | BLEU-3 | | METEOR | | ROUGE | | SPICE | | CIDEr | | S-BERT | | CapsMIX |
|---|---|---|---|---|---|---|---|---|---|---|---|---|---|---|---|---|---|
| | A | E | A | E | A | E | A | E | A | E | A | E | A | E | A | E | |
| | | | | | | | | *Finetuned* | | | | | | | | | | |
| BlindQA [2] | 0.2312 | 0.1847 | 0.0869 | 0.0550 | 0.0319 | 0.0198 | 0.1263 | 0.0930 | 0.2519 | 0.1806 | 0.2758 | 0.2447 | 0.0667 | 0.0265 | 0.4844 | 0.4153 | 2.7747 |
| UATT [53] | 0.2659 | 0.1936 | 0.1180 | 0.0645 | 0.0548 | 0.0223 | 0.1423 | 0.0868 | 0.3025 | 0.2200 | 0.3449 | 0.2595 | 0.0940 | 0.0713 | 0.5447 | 0.4750 | 3.2601 |
| HME [6] | 0.2640 | 0.1930 | 0.1238 | 0.0644 | 0.0556 | 0.0239 | 0.1457 | 0.0868 | 0.2998 | 0.2155 | 0.3115 | 0.1623 | 0.0944 | 0.0698 | 0.5341 | 0.4761 | 3.1207 |
| HGA [15] | 0.2674 | 0.2293 | 0.1337 | 0.0823 | 0.0620 | 0.0308 | 0.1703 | 0.1123 | 0.3186 | 0.2363 | 0.3734 | 0.2558 | 0.1972 | 0.0477 | 0.5765 | 0.4692 | **3.5628** |

**(c)**

Table 5: **OEAG Results** on our dataset. (a) **UD** split; (b) **PS** split; (c) **UN** split.

single static frame, instead of motion containing video clips. We believe these models can further improve their performance by incorporating explicit motion information. 2) Our CausalChaos dataset introduces the challenge of reasoning by linking scenes/shots. This is different from traditional temporal modeling, which is done within a scene. Scenes/shots involve an abrupt change in the scene. Traditional temporal modeling typically is geared toward smoother transitions; abrupt changes violate this condition. So while traditional temporal modeling does aid on our dataset, abrupt scene changes in our dataset poses a further challenge for VideoQA models which is not adequately addressed by traditional temporal models like 3DCNN feature extractors. We evaluated BLIP-2, VideoLLaMA and VideoChat2 on MCQA and Open-Ended Answer Generation (OEAG) task. We noticed that BLIP-2 model performed better on MCQA, while VideoChat2 and VideoLLaMA performed better on OEAG task. Notably, OEAG requires better language modeling than that required in MCQA task.

**Qualitative results.** We have provided the qualitative analysis of models' failure cases in the accompanying PowerPoint presentation for the following tasks/cases:

1. **Multi-Choice Question Answering (MCQA)**
2. **Open-Ended Answer Generation (OEAG)**
3. **Incorporating our data into real-world Video QA**

**Full results on OEAG** are presented in Table 5.

**Normalized performances on OEAG** are presented in Table 6. We normalize using upperbound of the individual metric, and then average across all the metrics. Upperbound of CapsMIX in this case would be 1.

### A.14 Discussion on cartoon physics

Cartoon physics often operates within its own set of rules and logic, which may differ from real-world physics but still maintain consistency within the cartoon's universe. These rules might include exaggerated movements, gravity-defying actions, and other fantastical elements that wouldn't occur in reality but are accepted within the context of the cartoon world. Despite the departure from

| Model | CapsMIX |
|-------|---------|
| *Finetuned* | |
| BlindQA | 0.1648 |
| UATT | 0.2032 |
| HME | 0.1848 |
| HGA | 0.2180 |
| BlindGPT-2 | 0.2993 |
| VisionGPT-2 | 0.3031 |
| VILA | **0.3475** |
| *Zero-shot* | |
| BLIP-2 | 0.0997 |
| Video-LLaMA | 0.1458 |
| VideoChat2 | **0.2084** |
| GPT-4o | 0.1719 |

Table 6: **Normalized OEAG results on our dataset (UD split).**

real-world physics, there is often an internal consistency to how these cartoon physics operate within their respective universes.

Despite the departure from real-world physics, humans/video understanding models can apply causal reasoning within the context of cartoon physics to predict the consequences of characters' actions. For example, if a character steps off a cliff, humans expect them to fall downwards due to gravity, even if the fall is exaggerated or prolonged for comedic effect. This consistency can allow video-understanding models to anticipate and understand the outcomes of actions within the cartoon world, facilitating their ability to follow the storyline and engage with the humor and narrative.

Furthermore, the consistency of cartoon physics enables humans to make logical connections between different events and understand the progression of the story. By recognizing patterns and understanding how actions lead to specific outcomes, video understanding models can engage in causal reasoning to predict future events and comprehend the logic of the cartoon universe.

### A.15 CausalConfusion incorrect/negative answer generation

Samples from the dataset created using Vanilla Hard Negative mining:

Q: Why did Tom dip his fingers in the ink?
Correct A: To draw a mouse hole on the wall.
Incorrect A(1): Tom wanted Jerry to mistake Tom's finger for a sausage.
Incorrect A(2): Tom was preparing to eat.
Incorrect A(3): Tom wanted to see if there was ink in the pen.
Incorrect A(4): Tom's hand was in pain.
Correct E: Tom was trying to trick Jerry by drawing a fake mouse hole on the wall.
Incorrect E(1): Tom's hand was in pain from hitting Jerry with the vase.
Incorrect E(2): Tom was excited to eat Jerry who was on Tom's plate.
Incorrect E(3): Tom thought there was no ink in the pen as the ink did not come out when Jerry pulled the pen.
Incorrect E(4): Tom wanted to trick Jerry to mistake Tom's finger for a sausage so that Tom could catch Jerry when Jerry tried to steal Tom's finger.

Q: Why did Tom climb onto the gate?
Correct A: The bull was charging towards Tom.
Incorrect A(1): Tom was trying to get away from Spike.
Incorrect A(2): Tom wanted to get to a higher point on the tree.
Incorrect A(3): because Tom heard barking sounds and was scared.
Incorrect A(4): Tom was trying to get away from Spike and Tyke.
Correct E: The bull was charging towards Tom so Tom climbed onto the gate to avoid getting hurt by the bull.
Incorrect E(1): because Jerry imitated Spike to bark at Tom to scare Tom into climbing up the tree.
Incorrect E(2): Tom was scared of Spike who was chasing Tom and climbed up the tree to get away from

Spike.
Incorrect E(3): Tom was dressed as a bird and wanted to climb higher on a tree to take off.
Incorrect E(4): Tom saw Spike and saw Tyke barking and wanted to get away from them.

Examples of Vanilla Hard Negatives vs. *CausalConfusion* Negatives:

Q: Why did Tom dip his fingers in the ink?
Correct A: To draw a mouse hole on the wall.
**Vanilla Hard Negatives**
Incorrect A(1): Tom wanted Jerry to mistake Tom's finger for a sausage.
Incorrect A(2): Tom was preparing to eat.
Incorrect A(3): Tom wanted to see if there was ink in the pen.
Incorrect A(4): Tom's hand was in pain.
**CausalConfusion version**
Incorrect A(1): To not draw a mouse hole on the wall.
Incorrect A(2): Tom was preparing to eat.
Incorrect A(3): Tom wanted to see if there was ink in the pen.
Incorrect A(4): Tom's hand was in pain.

Q: Why did Tom climb onto the gate?
Correct A: The bull was charging towards Tom.
**Vanilla Hard Negatives**
Incorrect A(1): Tom was trying to get away from Spike.
Incorrect A(2): Tom wanted to get to a higher point on the tree.
Incorrect A(3): because Tom heard barking sounds and was scared.
Incorrect A(4): Tom was trying to get away from Spike and Tyke.
**CausalConfusion version**
Incorrect A(1): The bull was not charging towards Tom.
Incorrect A(2): Tom was charging towards the bull.
Incorrect A(3): because Tom heard barking sounds and was scared.
Incorrect A(4): Tom was trying to get away from Spike and Tyke.

## A.16 CapsMIX extended details

However, we note that with such a wide range of metrics, it is difficult to get a comprehensive insight into models' performances & compare them. To address that, we introduce a comprehensive metric, termed Caps-MIX (Captioning Metrics Integration eXpert), which integrates all the previously mentioned scores after normalizing them to their theoretical best values. This **1)** makes it easier to compare models using a single number and **2)** combines the characteristics of individual metrics, each measuring performance from a unique perspective. We avoid using the WUPS score [28], as it is designed for single-word answers and is not suitable for our dataset's detailed responses.

## A.17 Negative societal impact

While our dataset has a positive attribute of being synthetic in nature. And as such, we do not suggest deploying models trained on our dataset in real-world applications. Causal reasoning models trained on real-world data can potentially be used to find out or estimate why people carried out actions. This, in-turn, can be used to deduce further actionable insights into people's behavior. This might justifiably be seen as an intrusion of privacy, especially, without consent. Thus, such systems shall not be deployed/used without the consent of all the parties involved. We suggest that this space should be regularized by governing bodies, and consent from the end-users, and parties being monitored is inevitable.

## A.18 Compute details

We used machine with following specifications: Intel(R) Xeon(R) W-2245 CPU@3.90GHz; 64GB RAM; 2x Nvidia A5000 24GB.

### A.19   Link to Dataset

We have included the following dataset files in the supplementary.

1. File containing all the annotations
2. Vanilla hard negative sets
3. CausalConfusion set

The dataset files are also publicly available at: [https://github.com/LUNAProject22/CausalChaos](https://github.com/LUNAProject22/CausalChaos).

### A.20   Adopting Image only VLMs for VideoQA

We adopt natively image-VLMs for VideoQA in the following ways. Adoption is dependent on the model. For example, for BLIP-2, we average the frame-level vision features to serve as our video-level feature. In Particular, we follow the standard practice and uniformly sample 16 frames from the video. For GPT4o, we use the following strategy: whilst they have yet to publicly release the video model it has been shown that it is able to summarize and understand videos by providing a sequence of images. As such, we pass in a sequence of 16 frames sampled uniformly from the video as the visual inputs.

