# OpenReview forum: "CausalChaos! Dataset for Comprehensive Causal Action Question Answering Over Longer Causal Chains Grounded in Dynamic Visual Scenes"
_NeurIPS.cc/2024/Datasets_and_Benchmarks_Track — NeurIPS 2024 Track Datasets and Benchmarks Poster_

### Official Review · Reviewer_nU6m · 2024-07-22
**Great and novel dataset with challenging tasks**

**Rating:** 8
**Confidence:** 3
**Clarity:** The paper is clearly written and grea…

**Review:**

The introduced video dataset is large and offers unique and novel challenges with annotated data. The authors put a lot of effort into ensuring high-quality and challenging datasets with detailed labeling information. Each of the design steps is logical and particularly challenges the drawbacks of state-of-the-art models in terms of reasoning. All experiments are carefully introduced and executed with very insightful results. The authors aimed at answering a wide range of possible questions and experiments. Seeing the novelty of the type of dataset and its potential impact on future evaluations of video/multi-modal models, this is great work.

**Strengths:**

- Great representation and writing.
- Good introduction, motivation, and detailed comparison with existing work.
- Huge dataset with a thoughtful annotation process leading to high-quality labels.
- Detailed experiments addressing a wide range of interesting questions and comparisons with the dataset.
- Fair discussion of potential impacts and limitations.

**Additional Feedback:**

Great paper, my main points and questions have been mentioned above. Maybe the authors can briefly comment on how one would use the dataset in practice given the requirement of the video material.

**Correctness:**

The dataset, labeling, and experimentation are thoughtfully conducted and evaluated.

**Documentation:**

Labeling processes and datasets have been discussed sufficiently in detail.

**Ethics:**

No ethic concerns.

**Limitations:**

The authors have a fair discussion of the limitations. However, more discussion on how one could use the dataset in practice due to potential licensing issues and potential asynchronicity issues could be helpful.

**Opportunities For Improvement:**

Don't have too many points here, just a few rather minor ones:

- When evaluating the reasoning (E), it is not entirely clear if this is based on free text (and then evaluated via similarity) or whether this is also offered as multiple choice answers. I might have missed this detail, but pointing this out more clearly and what this means for the insights of the experimental results can be helpful (e.g., how would choosing an answer first and then selecting the reasoning differ from first asking for reasoning and then selecting an answer).
- The results in Table 3 are difficult to interpret, i.e., what does this tell us about the dataset?
- It's unclear how the dataset can be used in practice due to potential licensing issues. How can it be ensured that the time-frames coincide between potentially different versions of the videos?

**Relation To Prior Work:**

Detailed and convincing discussion of related work.

**Summary And Contributions:**

The authors present a new benchmark dataset for causal reasoning within videos. Here, the authors use the Tom and Jerry cartoon as a base with annotated scenes of cause-effect relationships and the causal reasoning behind them. The dataset is utilized in various experiments to compare different existing reasoning models, but the authors also trained their own models and evaluated potential performance boosts when used as an additional data source.

---

> ### Author Rebuttal · Authors · 2024-08-17
>
> Thank you for your thoughtful and positive feedback on our work. We’re delighted that you appreciated the representation, writing, and the detailed introduction and motivation of the paper, as well as our efforts in creating and annotating a high-quality dataset. Your recognition of the comprehensive experiments, the discussion of potential impacts and limitations, and the novelty and challenge presented by our dataset is deeply appreciated. We value your insights and suggestions and are encouraged by your kind words as we continue to refine and advance our research. We hope the following clarifications can address the reviewer's concerns.
>
> **[R3] Are Explanations (E) offered as free text or multiple choice answers?**
>
> Thank you for your question, and we apologize for any confusion caused. In the Multiple Choice Question Answering (MCQA) task, the explanations are presented as multiple-choice answers. In contrast, the Open-Ended Answer Generation (OEAG) task evaluates models on generating free-text explanations. We appreciate this feedback and will clarify this distinction more explicitly in the revised manuscript.
>
>
> **[R3] What this means for the insights of the experimental results can be helpful (e.g., how would
> Scheme-1 (choosing an answer first and then selecting the reasoning) differ from Scheme-2 (first asking for reasoning
> and then selecting an answer))**
>
> Thank you for the insightful question. Both schemes have their merits. In this work, we are first choosing an answer and then selecting the reasoning like how humans would answer a question in everyday conversations. We observed choosing correct answer and reasoning to be more challenging than only selecting the correct answer for the models (L302-304). We might not be able to comment on how these two schemes differ as we only consider the first scheme in the paper. However, we hypothesize that a slightly different variant of the second scheme of coming up with a reasoning first can play an important role in the training phase and interpretability of the models. This is an area we are actively exploring in ongoing research.
>
>
> **[R3] The results in Table 3 are difficult to interpret, i.e., what does this tell us about the dataset?**
>
> This is an important question because of the following reasons. We believe that at the moment models are struggling to perform well on Open-Ended Answer Generation (OEAG), especially, when we take a deeper look at the full results that we provided in the Appendix. Scores on most of the metrics are actually low when we compare them with their upper bounds. These low performances could potentially be due to multiple reasons such as: 1) lack of joint vision-language modeling; 2) lack of modeling/generating long answers (or equivalently longer casual chains); 3) lack of standalone reasoning in order to generate answers rather than select answers. We believe a good, challenging dataset should be able to challenge these capabilities in VideoQA models. We believe our dataset offers these challenges because: 1) our dataset requires models to understand videos thoroughly in order to be able to answer the question (L157-186); 2) our dataset contains longer causal chains, which requires language modeling and generation capabilities (L187-194); our dataset is more challenging in terms of standalone reasoning (it involves multiple types of reasoning (207-213), longer causal chains embedded in dynamic scenes (L176-186)). We hope this answers the reviewer's question. Please let us know if we can provide any further explanation.
>
>
> **[R3] How can it be ensured that the time-frames coincide between potentially different versions of the videos?**
>
> Thank you for asking this practically important question. It is typical that different versions of cartoon series published by different publishers would contain publisher-specific credit rolls or branding segment appended in the very beginning---before the actual cartoon episode starts; or at the very end---after the episode ends; not during the episode (publisher logo and other small details might be found in the far corners of frames the during episode, but we believe that is not our concern(?)). This appendage can likely result in time-frames being off by a particular amount. We believe this reviewer’s concern (?). To mitigate this, we plan to use frame number immediately succeeding the Tom and Jerry’s credits (not the brand credits) for synchronization purposes. We will release synchronization frame indices. Different versions can then be synchronized based on these frame numbers. We will also offer case-specific troubleshooting support in case the synchronization method is not possible for any research groups. Please do let us know if we can further address your concern or if you have any further suggestions.
>
> **[R3] Maybe the authors can briefly comment on how one would use the dataset in practice given
> the requirement of the video material.**
>
> We believe the concern here is that generally video material could be freely accessed, while in our case video material will need to be bought, which incurs a cost. We have considered this aspect carefully. Typical cost for obtaining Tom and Jerry full series (all seasons and episodes) is about $60 USD. We believe this amount is well within the bounds of most research funding and research student scholarships. These funding sources also generally cover a wide range of expenses, including data acquisition. Moreover, if any research groups require a formal letter confirming that the video material needs to be purchased, we would be happy to provide it. Given these considerations, we do not anticipate that acquiring the video material will pose a significant barrier. We hope this answers your question; however, if we have misunderstood your concern, please feel free to clarify, and we will be glad to provide further details.

---

> > ### Author Response · Authors · 2024-08-17
> > **Newly Added Results**
> >
> > Additionally, we have evaluated some more baselines, including human baseline. We have attached results PDF for your consideration in the above Rebuttal section.

---

> > > ### Comment · Reviewer_nU6m · 2024-08-20
> > >
> > > I want to thank the authors for answering my questions and addressing my concerns. I maintain my (high) score and believe that this is a valuable contribution.

---

> > > > ### Author Response · Authors · 2024-08-21
> > > > **Thank You for Your Positive Feedback and Continued Support**
> > > >
> > > > Thank you very much for your continued support and positive feedback. We are truly grateful for your thoughtful comments and are pleased to know that our rebuttal effectively addressed your concerns. Your encouraging feedback and constructive input throughout the process have been instrumental and truly mean a lot to us.
> > > >
> > > > Thank you again for your time and support.

---

### Official Review · Reviewer_387i · 2024-07-26

**Rating:** 6
**Confidence:** 4

**Review:**

The CausalChaos! dataset paper presents an intriguing and valuable contribution to the field of causal video question answering, leveraging the "Tom and Jerry" cartoon series to create a challenging dataset that emphasizes longer causal chains and dynamic visual scenes. The inclusion of multi-level answers and hard incorrect answers is important.

However, there are areas for improvement. The quality of the explanations can be inconsistent and subjective, which might affect the reliability of the dataset, and there is no detail about how they were generated. Additionally, while the dataset shows potential for enhancing model performance on real-world datasets, the improvements observed are limited. Moreover, the evaluations conducted are somewhat restricted, as they do not include larger language models (LLMs) or vision-language models (VLMs) for the open-ended generation task. Addressing these aspects could significantly enhance the impact and applicability of the dataset.

**Strengths:**

1. Leverages the "Tom and Jerry" cartoon series for studying causal video question answering with complex causal chains and dynamic interactions. The video dataset is "out-of-distribution" from real-world scenes and has minimal amounts of text/audio.
2. Answers with explanations
3. Specifically created hard-negative answers.

**Additional Feedback:**

N/A

**Clarity:**

I don't understand the paragraph of L150-L156. There are a lot of vaguely defined words (e.g., nuanced perspective, deeper insights, character's decisions from personal factors...) in it and I think it should be clarified

**Correctness:**

The claims made in the submission mostly look correct. I think more investigations into the GPT-2 language-only performance should be performed.

**Documentation:**

Model evaluation code (especially the evaluation metrics) should be provded.

**Limitations:**

The dataset's explanations lack standardized quality control, making them subjective and potentially inconsistent, with insufficient details on the annotation protocol. Additionally, the evaluations exclude larger open-sourced language models (LLMs) and vision-language models (VLMs), limiting the understanding of the dataset's potential and practical impact.

**Opportunities For Improvement:**

First, the quality of explanations provided in the dataset does not seem to be fully controlled and can be subjective, which may affect the consistency and reliability of the annotations. The details of quality control and even the annotation protocol were not discussed in the paper. I personally suspect that free-form writing of answers and explanations makes it very hard to be consistent. For example, in which level of detail should the explanations be provided? If there is any commonsense knowledge required to understand the causal relationships between facts, what's the protocol to ensure that they are labeled? For example, in the first example in Figure 1 (Left), I do not see why the explanation is an "explanation" for the answer (I actually see the opposite, in the sense that the explanation should be the answer, and the current answer should be the explanation).

Second, while the new dataset is definitely very interesting, and challenging, it is unclear how significant they are. For example, if the goal is to improve model performance on real-world video QA applications (e.g., NextQA), the improvements shown there (Table 4) are very small. I can imagine that this paper will be useful for other scientific studies (e.g., Cognitive Science), but they are not discussed in the paper.

Third, the evaluations conducted in the study are somewhat restricted, as they do not include "real" open-sourced larger language models (LLMs) or vision-language models (VLMs) for the open-ended generation tasks. Incorporating these more recent, larger models (e.g., LLaMa, Qwen) would provide a more comprehensive understanding of the dataset's impact and potential. For example, it is very suspicious why GPT2 performs so well on this dataset, and it is interesting to see how more advanced LLMs or VLMs would perform on this task.

Furthermore, human performance should be reported, especially on the multiple choice split so that we can understand the hardness of the dataset.

Many figures (e.g., Figure 1 Left) should be made higher-resolution.

**Relation To Prior Work:**

Yes

**Summary And Contributions:**

The paper introduces the CausalChaos! dataset designed for causal video question answering (QA) based on the "Tom and Jerry" cartoon series. This dataset focuses on complex causal reasoning by presenting questions that require understanding of longer causal chains within dynamic visual scenes. Each question in the dataset comes with a multi-level answer, including both a direct response and a detailed explanation of the causal relationships

---

> ### Author Rebuttal · Authors · 2024-08-17
>
> We thank the reviewer for all the constructive suggestions and identifying the value of this dataset in other fields such as cognitive science. We hope the following clarifications can address the reviewer's concerns.
>
> **[R2] Addressing Annotation Protocol, Consistency, and Quality Control**
>
> We appreciate the reviewer’s concern regarding potential inconsistencies arising from free-form writing of answers and explanations, particularly in terms of the level of detail provided in explanations. Maintaining consistency is critical, especially when dealing with nuanced, multi-level causality, and we have taken deliberate steps to address this challenge.
>
> **Guiding Principle for Consistency.** Our dataset’s multi-level answer structure is specifically designed to explore multiple layers of causality, moving from direct, literal causes (Primary answers) to deeper, more contextual reasoning (Explanations). To maintain consistency across these levels, we employ a clear principle: the Explanation must extend the causal reasoning chain established by the Primary answer by introducing at least one additional causal link.
> This principle ensures that Explanations are not arbitrary or overly detailed but instead provide the next logical step in the causal chain. For example, if the Primary answer addresses an immediate reaction, the Explanation would explore the underlying motivations, intentions, or anticipated consequences, adding depth while adhering to a systematic structure.
>
> **Defining the Level of Detail in Explanations.** The level of detail in Explanations is determined by the need to capture a deeper level of causality without overwhelming the core causal relationship. Annotators are guided to expand upon the immediate cause by considering:
> * Broader context within the scene.
> * Characters’ thoughts, emotions, and intentions.
> * Potential consequences or future implications of actions.
>
> This approach ensures that the Explanations are neither excessively detailed nor superficial. By focusing on adding a single additional link to the causal chain, we maintain a consistent level of detail that systematically extends the reasoning without straying into irrelevant or redundant information.
>
> **Annotation and Quality Control Processes.** To further ensure consistency, we have implemented a structured annotation process combined with rigorous quality control:
> * **Annotator Training and Guidelines:** Annotators are trained using specific examples and guidelines that emphasize the principle of extending the causal chain. They are provided with clear instructions on how to transition from the Primary answer to the deeper Explanation while staying within a consistent level of detail.
> * **Multi-Stage Quality Checks:** We conduct a two-stage quality control process where annotators review episodes they did not initially annotate, followed by a second review phase where flagged issues are resolved through team discussions and consensus. This process helps ensure that Explanations maintain a consistent depth across the dataset while aligning with the intended multi-level causality structure.
>
> **Revision.** We recognize that some of these details may not have been fully articulated in the current version of the paper (L115-141). In response to this concern, we will revise the manuscript to include a more comprehensive description of the guiding principle for Explanations, along with clear examples.

---

> > ### Author Rebuttal · Authors · 2024-08-17
> >
> > **[R2] If there is any commonsense knowledge required to understand the causal relationships between facts, what's the protocol to ensure that they are labeled?**
> >
> > We appreciate the reviewer’s concern regarding the role of commonsense knowledge in understanding causal relationships and the protocols we use to ensure it is consistently labeled. In the following, we discuss them in detail to address the concerns.
> >
> > **The Role of Commonsense Knowledge.** Understanding causal relationships in complex scenes, such as those in Tom & Jerry episodes, often requires more than just visual cues—it requires the integration of commonsense knowledge. For example, understanding a character’s motivation to avoid danger or seek revenge may rely on widely understood commonsense concepts. We recognize that these implicit aspects of reasoning are crucial to accurately capturing multi-level causality.
> >
> > **Protocols for Labeling Commonsense Knowledge.** To address the potential need for commonsense reasoning, we have established specific protocols to ensure that it is consistently and systematically integrated into our annotations:
> > * **Annotator Training and Guidelines:** Annotators are explicitly trained to recognize when a causal relationship depends on commonsense knowledge that is not directly visible in the scene. The guidelines provide clear examples where commonsense knowledge plays a role, such as understanding fear, goal-directed behavior, or character motivations. Annotators are instructed to explicitly include this knowledge in the Explanation, ensuring that it is systematically captured rather than assumed.
> > * **Structured Annotation Principle:** Our guiding principle for consistency involves extending the causal reasoning chain beyond the Primary answer. When commonsense knowledge is needed to connect causal links, annotators are trained to articulate this reasoning within the Explanation. For instance, if a character’s action is driven by fear or anticipation (which requires commonsense understanding), the Explanation explicitly mentions this to avoid leaving any causal steps implicit.
> > * **Quality Control for Commonsense Knowledge:** The two-stage quality check process includes a focus on identifying gaps in causal reasoning that may be due to unarticulated commonsense knowledge. Quality checkers review Explanations to ensure that any required commonsense knowledge is explicitly labeled. If a causal relationship depends on widely understood facts or concepts (e.g., characters seek safety or pursue goals), the Explanation is reviewed to ensure this reasoning is clearly stated.
> > * **Resolving Ambiguities Through Consensus:** In cases where the role of commonsense knowledge is ambiguous, these instances are flagged during the quality check process and resolved through team discussions and consensus. This approach ensures that all annotators have a consistent understanding of when and how to incorporate commonsense knowledge into the labeled data.
> >
> > **Revision.** We recognize that the specifics of how commonsense knowledge is handled may not be fully detailed in the current version of the paper. We plan to revise the manuscript to include a more explicit description of our protocols, including how annotators are guided to recognize and label commonsense knowledge and how the quality control process ensures consistency.
> >
> > **[R2] “…in the first example in Figure 1 (Left), I do not see why the explanation is an "explanation" for the answer (I actually see the opposite, in the sense that the explanation should be the answer, and the current answer should be the explanation).”**
> >
> > ```Question: “Why did jerry put tom's tail in the sandwich?”```
> >
> > ```Answer (Level-1): "Jerry wanted Tom to bite his own tail."```
> >
> > Why this is first-level answer: This is the **first-level answer** because it directly describes the specific action and its **immediate consequence** without delving into the broader motive.
> >
> > ```Explanation (Level-2): "Jerry wanted to inflict pain on Tom."```
> >
> > Why this is second-level answer: This is the **second-level answer** because it delves into Jerry's **broader/deeper intention**, suggesting an underlying motive of causing pain.
> >
> > We hope it would be more clear how our annotations delve deeper as they go from Primary Answer to Second-level Explanation. We also suggest considering Subsection A.9 in the Appendix. Please let us know if we can provide with further explanation or clarification.

---

> > ### Author Rebuttal · Authors · 2024-08-17
> >
> > **[R2] Clearing the confusion surrounding Table 3 (and GPT-2's relatively better performance)**
> >
> > We are extremely sorry for the confusion. We believe some confusion is stemming from the divider line placed between VisionGPT-2 and BLIP-2. Please allow us to clarify it. We meant the divider line to differentiate between finetuned and zeroshot-tested models; but unfortunately, we forgot to mention the word ‘Zeroshot’ above the divider line. Without the ‘Zeroshot’ heading, it may likely be misconstrued as differentiating between traditional models and recent VLMs.
> > All the models above the line (including the GPT-2-based models) are finetuned on our dataset. While models below the divider line (BLIP-2, Video-LLaMa, VideoChat2, GPT-4o) are not finetuned and tested zeroshot on our dataset as mentioned in L329. We believe now this would clarify why GPT-2-based models are performing better than the recent VLMs. To further clarify, GPT-2-based models without finetuning perform significantly worse (L343-344). We have fixed this typo and will update the paper accordingly. We will address further concerns that might be related to this in the following.
> >
> > **[R2] Selection of Vision-Language Models (VLMs) and the Evaluation of Strong Baselines**
> >
> > Thank you for raising this important point. We apologize for any confusion that may have arisen, particularly due to the absence of the “Zero-Shot” label above the divider line in Table 3. We understand that this may have led to an impression that we did not consider some of the latest and strongest VLMs as baselines.
> >
> > We would like to clarify that the VLMs we chose for evaluation were carefully selected based on their demonstrated strong performance in video understanding tasks. Specifically, the models we included—GPT-4o, VideoChat2, VideoLLaMA, and VILA (newly added)—are recognized for their high performance across multiple video understanding benchmarks and leaderboards, such as MLVUBench (reference: MLVUBench Leaderboard) and MVBench (CVPR 2024) [21].
> >
> > In terms of recency, the models we evaluated are among the latest available:
> >   *  VideoChat2 was published in CVPR 2024 (held in June 2024).
> >   *  GPT-4o was released in mid-May 2024, just two weeks before the NeurIPS submission deadline.
> >   *  VILA (now added) was published in CVPR 2024. We also further finetune this model on our dataset and report its performance.
> >
> > Furthermore, the models we selected represent improvements upon earlier strong baselines. For example:
> >   *  VideoLLaMA builds upon LLaMA (LLaMA is suggested by you(R2)).
> >   *  VILA (newly added) VLM geared for video understanding that builds upon LLaVA (LLaVA was suggested by R1(sN8y)).
> >
> > By including these models, we believe we have evaluated some of the most advanced and top-performing VLMs available, ensuring a fair and competitive comparison.
> >
> > We hope this clarifies our selection process and demonstrates that our study includes robust and representative baselines.
> >
> > **[R2] Provide Human Baseline.**
> >
> > Thank you for the suggestion. We had obtained a human baseline on our dataset (MCQA) based on five human subjects (fluent in English language and education at least studying in undergrad level or higher) on a subset of our dataset consisting of 100 randomly chosen samples. We have presented the human baseline in the attached PDF file. We will also add these results in the main paper. We acknowledge the importance of including this baseline in the main paper for better context and comparison. We appreciate your feedback and believe that incorporating this will strengthen the overall presentation of our work.
> >
> > **[R2] Many figures (e.g., Figure 1 Left) should be made higher-resolution.**
> >
> > Thank you for the helpful suggestion. We agree that higher-resolution figures would improve the clarity of the visual information presented. In the revised version of our paper, we will enlarge and enhance the resolution of the relevant figures, including Figure 1 (Left). Additionally, we will provide further enlarged figures, along with video samples, in the Appendix for better illustration. We appreciate your feedback and believe this improvement will significantly enhance the presentation quality of our work.
> >
> > **[R2] Model evaluation code (especially the evaluation metrics) should be provided.**
> >
> > Thank you for this important suggestion. We agree that providing the evaluation code, especially for the metrics, is crucial for reproducibility and transparency. We are pleased to inform that the initial release of the code is now publicly available at: https://github.com/LUNAProject22/CausalChaos.

---

> > ### Author Rebuttal · Authors · 2024-08-17
> >
> > **[R2] Modest improvements when integrating with real-world dataset.**
> >
> > We sincerely appreciate the reviewer's thoughtful feedback and their attention to the results of our work. We understand that the mild nature of the improvements observed may raise concerns, and we would like to provide some additional context to clarify why we believe these results are still meaningful contributions to the field.
> >
> > * **1. Importance of Cross-Domain Data Integration:**
> > Our primary goal was to explore the feasibility of integrating cartoon-derived datasets with real-world data for causal video question answering. Given the significant domain gap between stylized cartoon content and real-world videos, avoiding a performance drop is, in itself, a noteworthy achievement. Cross-domain learning often carries a high risk of negative transfer, where irrelevant or misleading patterns from one domain can harm performance in another. The fact that the cartoon dataset did not degrade performance across multiple models is a good indication that it introduces valuable, complementary information without introducing noise.
> >
> > * **2. Consistency Across Multiple Models:**
> > We also tested this approach across multiple models, and the consistent stability across different architectures further reinforces the idea that the findings are not tied to a specific model design. We see this as an indication that the benefits of integrating this type of data are broadly applicable, highlighting the dataset’s quality and relevance despite its stylized nature and providing a foundation for further research in this area.
> >
> > * **3. Broader Implications of Data Diversity:**
> > While the improvements are admittedly modest, we believe they are still valuable. They suggest that even stylized or abstract datasets can contribute positively when combined with real-world data. We also find it interesting that the cartoon dataset, which contains only why questions, led to improvements across both why and non-why question types. This suggests that the model’s enhanced causal reasoning skills and learnt representations  generalize beyond the specific questions seen during training, potentially improving overall video comprehension.
> >
> > * **4. Potential for Future Exploration:**
> > We agree that more substantial improvements would be better, but we believe that our findings highlight the potential for integrating diverse and non-traditional datasets without risking performance degradation. We hope this work can serve as a starting point for further exploration into the benefits of using stylized or synthetic data in tasks that involve causal reasoning or video understanding, especially when real-world data is limited.
> >
> > **Conclusion:**
> > In conclusion, we see our study as a first step in demonstrating that integrating cartoon datasets with real-world data can be done safely, without harming performance, and with the potential for modest improvements. We acknowledge that the improvements are mild, but we believe the consistency of results across models and question types indicates that this approach has value and warrants further investigation. We appreciate the reviewers’ careful consideration of our work and hope this additional context clarifies our perspective on the contributions and future implications of our findings.
> >
> >
> > **[R2] Potential relevance of the paper to other scientific studies (e.g., Cognitive Science) is not discussed.**
> >
> > Thank you for highlighting this valuable point. We agree that our work has potential applications beyond the immediate scope of video understanding, particularly in fields such as Cognitive Science, where the study of causal reasoning, perception, and decision-making is central. We realize that these broader interdisciplinary impacts were not explicitly discussed in the current version of the paper.
> >
> > In the revised manuscript, we will include a section discussing how our dataset and findings could be relevant to Cognitive Science and other related fields. Specifically, we will explore how our dataset could be used to study cognitive processes like causal inference, theory of mind, visual storytelling, child development research/applications. By addressing these connections, we aim to emphasize the broader scientific utility of our work.
> >
> > We truly appreciate your insightful suggestion and will incorporate this discussion to enrich the paper’s interdisciplinary relevance.

---

> > ### Author Rebuttal · Authors · 2024-08-17
> >
> > **[R2]  I don’t understand the paragraph in Lines 150-156. There are a lot of vaguely defined words (e.g., “nuanced perspective,” “deeper insights,” “character’s decisions from personal factors”) in it, and I think it should be clarified.**
> >
> > Thank you for your feedback. We recognize that some terms in this paragraph may have been unclear and could benefit from more precise explanations. Here are clearer definitions and explanations:
> > * **Nuanced Perspective leading to Deeper Insights:** In this context, these terms refer to the ability to consider and differentiate between multiple potential causes or contributing factors when analyzing a character’s actions. A nuanced perspective involves exploring not just the obvious or immediate reasons behind a decision but also underlying influences such as long-term motivations, social pressures, or personal history. By considering these multiple layers of causality, we gain deeper insights into the character’s behavior and decision-making process.
> > * **Character’s Decisions from Personal Factors:** This refers to how decisions are shaped by a character’s unique background, experiences, and psychological traits. For instance, a character might make a decision based on deeply ingrained fears, past experiences, or personal values. Understanding these internal drivers is key to evaluating a character’s actions beyond just external situational factors.
> >
> > We have also further elaborated on these concepts in Appendix Section A.9. In the revised manuscript, we have combined these concepts and provided more concrete explanations and examples to make the discussion clearer. We hope this addresses your concern and improves the clarity of the manuscript.

---

### Official Review · Reviewer_sN8y · 2024-08-01
**Review of CausalChaos!**

**Rating:** 7
**Confidence:** 4
**Clarity:** This paper is well-written and easy t…

**Review:**

This paper discovers the limitations of the existing video causal QA datasets and builds a high-quality dataset along with the training and evaluation pipelines. The proposed dataset can be seen as a powerful resolution of a lack of deep video causal datasets.
The strengths of this work can be summarized as:
1. The authors proposed a high-quality video causal dataset with a larger causal depth and multiple causal types.
2. The dataset is annotated by human annotators, ensuring the quality of the dataset. In the era of large models, this is particularly valuable.
3. The authors take some important features, such as the design of DSL into consideration. These features are important for AI to understand the real distribution of the world but remain under-explored in the field of video understanding.

However, I also have a few of questions.
1. I have noticed that the authors select some VLMs such as GPT-4o[1], and BLIP-2[2] as baseline models. But as far as I know, these 2 models[1-2] cannot process video directly. I failed to find the training and evaluation details of Image-only VLMs[1-2]. I wonder if I missed some information about the details. If image-only VLMs can be trained and tested, there exist quantities of open-source VLMs such as LLaVA[3], phi-3v[6], mPLUG-Owl[4], and Qwen-VL[5] .etc which can be used as baselines.
2. I read the detailed evaluation results of OEAG and found the traditional models (BlindGPT-2, VisionGPT-2) outperform over video understanding models at the CapsMIX benchmark. Could this be due to overfitting to the linguistic patterns in the dataset? Have you ever analyzed the failed cases of traditional and VLM models over the CausalChaos! datasets?
3. I failed to find the detailed results of GPT-4o[1], can you release them?
4. I failed to find the evaluation of human performance, namely how well human beings can perform on CausalChaos!. Knowing this can help to find the upper boundary of the model performance.
5. The dataset is constructed from cartoon videos, and the authors claim it can help with real-world cases. However, in my opinion, the improvements are not clear enough and the baselines do not contain the VLMs. Can you provide further analysis of how this dataset can bring benefits to real-world applications? And could the bias between cartoons and the real world bring negative effects on model performance?

To conclude, this paper addresses the raised limitations and introduces novel features. If my questions are answered, I will consider increasing my rating.

References

[1]OpenAI. Hello gpt-4o. https://openai.com/index/hello-gpt-4o/, 2024. [Online; accessed 478 31-May-2024].

[2]Junnan Li, Dongxu Li, Silvio Savarese, and Steven Hoi. BLIP-2: bootstrapping language-image pre-training with frozen image encoders and large language models. In ICML, 2023.

[3]Liu, Haotian, et al. "Visual instruction tuning." Advances in neural information processing systems 36 (2024).

[4]Ye, Qinghao, et al. "mplug-owl: Modularization empowers large language models with multimodality." arXiv preprint arXiv:2304.14178 (2023).

[5]Bai, Jinze, et al. "Qwen-vl: A versatile vision-language model for understanding, localization, text reading, and beyond." (2023).

[6]Abdin, Marah, et al. "Phi-3 technical report: A highly capable language model locally on your phone." arXiv preprint arXiv:2404.14219 (2024).

**Strengths:**

The strengths are the same as claimed in the "review" part.
1. The authors proposed a high-quality video causal dataset with a larger causal depth and multiple causal types.
2. The dataset is annotated by human annotators, ensuring the quality of the dataset. In the era of large models, this is particularly valuable.
3. The authors take some important features, such as the design of DSL into consideration. These features are important for AI to understand the real distribution of the world but remain under-explored in the field of video understanding.

**Additional Feedback:**

N/A

**Correctness:**

The claims are correctly made, and the evaluation methods and experiment design are claimed clearly. However, the evaluation details about VLMs and part of VLMs' experiment results remain unclear.

**Documentation:**

The data collecting and dataset constructing procedures are clearly stated.

**Limitations:**

The limitations are the same as claimed in the "review" part. Please refer to the ‘review’ section.

**Opportunities For Improvement:**

Please refer to the ‘review’ section.

**Relation To Prior Work:**

The relation between this work and prior works is discussed clearly.

**Summary And Contributions:**

This paper proposes a dataset for comprehensive video causal QA tasks. The authors notice the limitations among existing video causal QA datasets and make a nice attempt to resolve the limitations.
The noticed limitations of existing works are:
1. surface-level understanding and lack of complexity reasoning.
2. involve more simple word substitution in the QA pairs, rather than causal reasoning
3. limited scope
4. lack of precise
5. lack of hard negatives

And respectively, the main contribution of this work can be summarized as follows:

1. This work formulates thought-provoking questions, forming a much longer causal chain, which is subsequently challenging.
2. This work includes scene and shot changes of the videos, which are under-researched among related works.
3. This work contributes a dataset from animation, in which the granularity of move is different from real-world videos. This requires a more specified understanding ability of models.
4. The proposed dataset demands a diverse range of reasoning skills.

---

> ### Author Rebuttal · Authors · 2024-08-17
>
> Thank you very much for your detailed and insightful feedback. We are thrilled that you found our proposed video causal dataset to be high-quality, especially regarding its depth and diversity of causal types, and that you recognize the value of human annotation in enhancing dataset quality in the era of large models. Your acknowledgment of the importance of features like DSL design, which are under-explored yet crucial for AI’s understanding of real-world distributions, is particularly encouraging. We appreciate your recognition of the novel challenges posed by our dataset, including extended causal chains, scene transitions, and the unique demands of animation-based video data. Your feedback motivates us to continue exploring these under-researched areas and refining our contributions to the field of video understanding. We hope the following clarifications can address the reviewer's concerns.
>
> **[R1] We would like to start by clearing the confusion surrounding Table 3 (and GPT-2's relatively better performance)**
>
> We are extremely sorry for the confusion. We believe some confusion is stemming from the divider line placed between VisionGPT-2 and BLIP-2. Please allow us to clarify it. We meant the divider line to differentiate between finetuned and zeroshot-tested models; but unfortunately, we forgot to mention the word ‘Zeroshot’ above the divider line. Without the ‘Zeroshot’ heading, it may likely be misconstrued as differentiating between traditional models and recent VLMs.
> All the models above the line (including the GPT-2-based models) are finetuned on our dataset. While models below the divider line (BLIP-2, Video-LLaMa, VideoChat2, GPT-4o) are not finetuned and tested zeroshot on our dataset as mentioned in L329. We believe now this would clarify why GPT-2-based models are performing better than the recent VLMs. To further clarify, GPT-2-based models without finetuning perform significantly worse (L343-344). We have fixed this typo and will update the paper accordingly. We will address further concerns that might be related to this in the following.
>
> **[R1] Selection of Vision-Language Models (VLMs) and the Evaluation of Strong Baselines**
>
> Thank you for raising this important point. We apologize for any confusion that may have arisen, particularly due to the absence of the “Zero-Shot” label above the divider line in Table 3. We understand that this may have led to an impression that we did not consider some of the latest and strongest VLMs as baselines.
>
> We would like to clarify that the VLMs we chose for evaluation were carefully selected based on their demonstrated strong performance in video understanding tasks. Specifically, the models we included—GPT-4o, VideoChat2, VideoLLaMA, and VILA (newly added)—are recognized for their high performance across multiple video understanding benchmarks and leaderboards, such as MLVUBench (reference: MLVUBench Leaderboard) and MVBench (CVPR 2024) [21].
>
> In terms of recency, the models we evaluated are among the latest available:
>   *  VideoChat2 was published in CVPR 2024 (held in June 2024).
>   *  GPT-4o was released in mid-May 2024, just two weeks before the NeurIPS submission deadline.
>   *  VILA (now added) was published in CVPR 2024. We also further finetune this model on our dataset and report its performance.
>
> Furthermore, the models we selected represent improvements upon earlier strong baselines. For example:
>   *  VideoLLaMA builds upon LLaMA (LLaMA is suggested by R2 (387i)).
>   *  VILA (newly added) VLM geared for video understanding that builds upon LLaVA (LLaVA was suggested by you (R1)).
>
> By including these models, we believe we have evaluated some of the most advanced and top-performing VLMs available, ensuring a fair and competitive comparison.
>
> We hope this clarifies our selection process and demonstrates that our study includes robust and representative baselines.
>
>
> **[R1] How do you use natively image only VLMs for VideoQA?**
>
> Thank you for this question. We adopt natively image-VLMs for VideoQA in the following ways. Adoption is dependent on the model. For example, for BLIP-2, we average the frame-level vision features to serve as our video-level feature. In Particular, we follow the standard practice and uniformly sample 16 frames from the video. For GPT4o, we use the following strategy: whilst they have yet to publicly release the video model it has been shown that it is able to summarize and understand videos by providing a sequence of images. As such, we pass in a sequence of 16 frames sampled uniformly from the video as the visual inputs. We will add this information along with further details such as prompts in the Appendix. Furthermore, it will be included in our codebase to be released, so future work will be able to use them. Please let us know if we can provide further information or clarification on this.

---

> > ### Author Rebuttal · Authors · 2024-08-17
> >
> > **[R1] On OEAG, could GPT-2 models be performing better than traditional video understanding models because they might be overfitting on the linguistic patterns in the dataset?**
> >
> > Thank you for this insightful observation. We believe there are two key aspects to consider when addressing this concern.
> >
> > First, GPT-2 models have inherent language generation capabilities, which give them an advantage in producing linguistically coherent outputs, even if those outputs do not necessarily reflect correct answers. This could explain why GPT-2 models seem to perform relatively better than traditional video understanding models, which are not optimized for language generation tasks.
> >
> > Second, while GPT-2 models appear to perform better in comparison, it’s important to emphasize that this is only relative. When examining the absolute performance scores presented in the full results (see Appendix), it becomes clear that GPT-2’s performance is still quite limited and leaves significant room for improvement. This suggests that, despite the relative advantage, GPT-2 may indeed be struggling with this task and could be underperforming overall.
> >
> > To reinforce this understanding, we will emphasize in the paper that GPT-2’s seemingly better performance in comparison to traditional models does not imply that it is performing well in absolute terms. We hope this addresses your concern, and we will incorporate these clarifications in the revised manuscript.
> >
> > **[R1] Have you analyzed the failed cases of traditional and VLM models over the CausalChaos! datasets?**
> >
> > Thank you for your question. Yes, we have conducted a thorough analysis of the failure cases of both traditional models and VLMs on the CausalChaos! datasets. The details of this analysis, including specific examples of failure modes, are provided in the Appendix Part 2 (available in both PowerPoint and PDF formats).
> >
> > Due to file size limitations on the OpenReview platform (50MB), we obtained permission from the Program Chair to provide the PowerPoint version (which includes videos) via a Google Drive link. We apologize for any inconvenience this might cause, but this format allowed us to present the analysis with richer visual examples.
> >
> > Our analysis of these failure cases directly informed key insights presented in the paper, particularly those discussed in Lines 279-285. This analysis helped us better understand the strengths and limitations of both traditional models and VLMs, offering more informed conclusions about their performance.
> >
> > We hope this clarifies our approach, and we are happy to address any further questions or concerns.
> >
> > **[R1] Provide the OEAG full results of GPT4o.**
> >
> > We sincerely apologize and have provided the full results in the attached PDF file. We will further include these results in the Appendix as well.
> >
> > **[R1] Provide the Human baseline on CausalChaos! dataset.**
> >
> > Thank you for the suggestion. We had obtained a human baseline on our dataset (MCQA) based on five human subjects (fluent in English language and education at least studying in undergrad level or higher) on a subset of our dataset consisting of 100 randomly chosen samples. We have presented the human baseline in the attached PDF file. We will also add these results in the main paper. We acknowledge the importance of including this baseline in the main paper for better context and comparison. We appreciate your feedback and believe that incorporating this will strengthen the overall presentation of our work.
> >
> > **[R1] Can you provide further analysis of how this dataset can bring benefits to real-world applications?**
> >
> > Thank you for your question. We have provided visual qualitative results/analysis in the Appendix Part 2 (PowerPoint version contains videos). Hopefully, this answers your question. Please let us know if you might be interested in any further analysis or information.
> >
> > **[R1] Evaluate VLM for real-world transfer experiment.**
> >
> > Thank you for the pointer. We have finetuned and evaluated the VILA1.5 model [CVPR 2024] for the transfer experiment. The results are provided in the attached PDF file. We will also add results to the main paper.

---

> > ### Author Rebuttal · Authors · 2024-08-17
> >
> > **[R1] Modest improvements when integrating with real-world dataset.**
> >
> > We sincerely appreciate the reviewer's thoughtful feedback and their attention to the results of our work. We understand that the mild nature of the improvements observed may raise concerns, and we would like to provide some additional context to clarify why we believe these results are still meaningful contributions to the field.
> >
> > * **1. Importance of Cross-Domain Data Integration:**
> > Our primary goal was to explore the feasibility of integrating cartoon-derived datasets with real-world data for causal video question answering. Given the significant domain gap between stylized cartoon content and real-world videos, avoiding a performance drop is, in itself, a noteworthy achievement. Cross-domain learning often carries a high risk of negative transfer, where irrelevant or misleading patterns from one domain can harm performance in another. The fact that the cartoon dataset did not degrade performance across multiple models is a good indication that it introduces valuable, complementary information without introducing noise.
> >
> > * **2. Consistency Across Multiple Models:**
> > We also tested this approach across multiple models, and the consistent stability across different architectures further reinforces the idea that the findings are not tied to a specific model design. We see this as an indication that the benefits of integrating this type of data are broadly applicable, highlighting the dataset’s quality and relevance despite its stylized nature and providing a foundation for further research in this area.
> >
> > * **3. Broader Implications of Data Diversity:**
> > While the improvements are admittedly modest, we believe they are still valuable. They suggest that even stylized or abstract datasets can contribute positively when combined with real-world data. We also find it interesting that the cartoon dataset, which contains only why questions, led to improvements across both why and non-why question types. This suggests that the model’s enhanced causal reasoning skills and learnt representations  generalize beyond the specific questions seen during training, potentially improving overall video comprehension.
> >
> > * **4. Potential for Future Exploration:**
> > We agree that more substantial improvements would be better, but we believe that our findings highlight the potential for integrating diverse and non-traditional datasets without risking performance degradation. We hope this work can serve as a starting point for further exploration into the benefits of using stylized or synthetic data in tasks that involve causal reasoning or video understanding, especially when real-world data is limited.
> >
> > **Conclusion:**
> > In conclusion, we see our study as a first step in demonstrating that integrating cartoon datasets with real-world data can be done safely, without harming performance, and with the potential for modest improvements. We acknowledge that the improvements are mild, but we believe the consistency of results across models and question types indicates that this approach has value and warrants further investigation. We appreciate the reviewers’ careful consideration of our work and hope this additional context clarifies our perspective on the contributions and future implications of our findings.
> >
> >
> > **[R1] Could the bias between cartoons and the real world bring negative effects on model performance?**
> >
> > We hope our previous answer covers this aspect. Please let us know if we can provide any further information or clarification.

---

> > > ### Comment · Reviewer_sN8y · 2024-08-19
> > > **Comment**
> > >
> > > Thank you for your kind and detailed response. I am delighted that most of my questions have been thoroughly addressed. Honestly, the additional information provided has offered me profound insights into this field. To summarize, the dataset proposed by the authors effectively addresses the identified weaknesses and significantly enhances the evaluation baselines for the model’s causal VQA capabilities. This represents a challenging yet vital advancement that offers substantial opportunities for enhancement in multimodal video models. Once again, I appreciate your thoughtful reply, and I will adjust my rating to a 7.

---

> > > > ### Author Response · Authors · 2024-08-19
> > > > **Thank You for Your Positive Feedback and Support**
> > > >
> > > > Thank you very much for your positive and encouraging feedback. It means a lot to us. We are delighted to hear that our response provided valuable insights and addressed your concerns effectively. We greatly appreciate your recognition of the significance of our dataset.
> > > >
> > > > Your constructive comments and thoughtful engagement throughout the review process have been invaluable in helping us improve our work. It is truly rewarding to hear that you found our additional information helpful and that you recognize the potential impact of our research. We are sincerely grateful for your consideration in revising your rating and for your supportive comments.
> > > >
> > > > Thank you once again for your time and consideration.

---

### Author Rebuttal · Authors · 2024-08-17

We would like to thank all reviewers for their valuable comments. We hope our responses have adequately addressed their previous concerns; and we look forward to addressing any further concerns. We take this as a great opportunity to improve our work and shall be grateful for any additional feedback you could give us.

(For easier reading, we have used Reviewer R1 for sN8y; R2 for 387i; R3 for nU6m)

In summary, we addressed some of the main concerns as follows. Detailed answers provided in Reviewers' sections.

* **Evaluate Human Baseline:** We have included the Human Baseline Results in the attached PDF.

* **Were strong VLMs evaluated on your dataset?** Yes, we have evaluated some of the strongest and latest VLMs (**VideoChat2** [CVPR 2024], **VideoLLaMA** builds upon LLaMA, **GPT4o** (released May 2024), **VILA1.5** [CVPR 2024] builds upon LLaVA). Updated result: we also **finetuned VILA1.5** on our dataset and included the results.

* **Addressing Annotation Protocol, Consistency, and Quality Control:** We had used a guiding principle where Explanations extend the Primary Answer causal chain by introducing one additional causal link, ensuring systematic and relevant depth. Annotators were trained with specific guidelines, and a multi-stage quality control process ensures uniformity. The manuscript will be revised to better articulate these methods and provide clear examples. Elaborated answer in the following in the Reviewers' sections.

* **Misunderstanding surrounding GPT-2 models:** GPT-2 performance was relatively higher because it was finetuned on our dataset, while other models were not---we had mentioned this in the text, but inadvertently forgot to mention in Table 3. Updated result: We finetuned a latest VLM---VILA1.5 and show that it can outperform GPT-2 to dispel further doubts.

* **Modest improvements when integrating with real-world dataset.** We discussed why improvements, although admittedly modest are important, and how it can help address the domain gap. Updated result: we also provided **VLM baseline---VILA1.5** and show the mild improvements hold.

* **How to synchronize videos from potentially different publishers?** We discussed our strategy for synchronizing videos.

* **Addressing the accessibility and cost of required video material for dataset usage:** We believe the cost of acquiring the full package of video material (~$60 USD) is manageable within typical research budgets and offer support in providing formal letters if needed.

---

### Decision · Program_Chairs · 2024-09-26

**Decision:**

Accept (Poster)

**Comment:**

The authors notice the limitations among existing video causal QA datasets and make a nice attempt to resolve the limitations.
Concretely, this paper contributes a high-quality video causal dataset based the 'Tom and Jerry' cartoon series. The introduced dataset emphasizes longer causal chains and dynamic visual scenes, which requires a more specified understanding ability of models.

All reviewers lean towards accepting this paper and have identified the following strengths:

- Great presentation and clear motivation
- Large-scale video causal dataset with a thoughtful annotation process leading to high-quality labels
- The dataset proposed by the authors effectively addresses the identified weaknesses and significantly enhances the evaluation baselines